# Performance of wind assessment datasets in United States coastal areas

Lindsay M. Sheridan[1], Jiali Wang[2], Caroline Draxl[3], Nicola Bodini[4], Caleb Phillips[4], Dmitry Duplyakin[4], Heidi Tinnesand[5], Raj K. Rai[1], Julia E. Flaherty[1], Larry K. Berg[1], Chunyong Jung[2], Ethan Young[4], and Rao Kotamarthi[2]

[1]Pacific Northwest National Laboratory, Richland, Washington, 99354, United States
[2]Argonne National Laboratory, Lemont, Illinois, 60439, United States
[3]Electric Power Research Institute, Washington DC, 20005, United States
[4]National Renewable Energy Laboratory, Golden, Colorado, 80401, United States
[5]One Energy, Findlay, Ohio, 45840, United States

*Correspondence to*: Lindsay M. Sheridan (lindsay.sheridan@pnnl.gov)

**Abstract.** The atmospheric dynamics that occur near the intersection of land and water offer exciting and challenging opportunities for wind energy deployment in coastal locations. New models and tools are continually being developed in support of wind resource assessment, and three recent products are explored in this work for their performance in representing characteristics of the wind resource at coastal locations: the Global Wind Atlas 3 (GWA3), the 2023 National Offshore Wind data set (NOW-23), and the wind climate simulations that are a component of the Wind Integration National Dataset (WIND) Toolkit Long-term Ensemble Dataset (WTK-LED Climate). These relatively new products are freely available and user-friendly so that anyone from a utility-scale developer to a resident or business owner can evaluate the potential for wind energy generation at their location of interest.

The validations in this work provide guidance on the accuracy of wind resource assessments for coastal customers interested in installing small or midsize wind turbines ($\leq$ 1 MW in capacity) to support energy needs at the residential, business, or community scale, such as the island and remotely located participants of the U.S. Department of Energy's Energy Transitions Initiative Partnership Project. At 23 coastal locations across the United States, dataset performance varies according to different evaluation metrics. All three recent datasets tend to overestimate the observed coastal wind resource. GWA3 produces the smallest annual average wind speed relative errors, whereas WTK-LED Climate is in best agreement in terms of representing diurnal wind speed cycles. NOW-23 is the highest performing of the datasets for representing seasonal and inter-annual trends in the coastal wind resource. While GWA3 and WTK-LED Climate are relatively insensitive to the dataset output heights selected for wind resource assessment at small and midsize wind turbine hub heights (20 m – 60 m), significant variation in the NOW-23 representation of wind shear across the wind profile in the lowest 100 m of the atmosphere leads to notable differences in wind speed estimates according to the dataset output heights selected for evaluation. GWA3 exhibits challenges in representation of observed wind speed diurnal cycles at small and midsize turbine hub heights, likely due to the dataset's consistent treatment of hourly wind speed trends regardless of altitude.

# 1 Introduction

Coastal communities, particularly those on islands or in remote locations, have unique characteristics that impact the energy technologies available to support them, including exposure to extreme weather, proximity to sea spray, and transportation availability for fuels or components. While much of the current wind energy focus in the United States is on the developing offshore wind market in support of large-scale generation and distribution, residents and community planners in coastal areas have opportunities to explore the potential of wind energy at a local level to power homes, businesses, farms, and facilities.

The U.S. Department of Energy's (DOE's) Energy Transitions Initiative Partnership Project (DOE, 2024a) collaborates with island and remote communities to advance their energy portfolios and increase their energy resilience, and distributed wind energy is frequently considered for its potential to address both goals.

    The unique atmospheric dynamics that exist near the intersection of land and water provide exciting and challenging opportunities for wind energy deployment in coastal locations. Sea breeze circulations and low-level jets are frequently

observed along U.S. coastlines (Parish, 2000; Bao et al., 2023; McCabe and Freedman, 2023). Such atmospheric phenomena impact the magnitude, timing, and forecast success in representing wind resource availability for wind energy generation (Storm et al., 2008; Carrasco-Díaz et al., 2011). As an example of the opportunities and challenges for coastal wind energy deployment, three small wind turbines at Jennette's Pier in North Carolina were generating about 20% of the pier's needs until Hurricane Dorian damaged the blades in 2019. For the pier, the opportunities outweighed the challenges and new wind

turbine systems are planned for installation (Brindley Beach, 2022).

    The sparsity of publicly-available observation data to support comprehensive wind resource assessment has driven the development of a variety of models, datasets, and tools over the last decade. Three datasets that were specifically developed to support the wind energy industry are evaluated in this work. The Global Wind Atlas 3 (GWA3) is a high-resolution wind database developed by the Technical University of Denmark (DTU) and the World Bank Group to support policymakers,

planners, and investors in identifying areas of high wind resource for wind energy generation at locations across the world (Davis et al., 2023). The 2023 National Offshore Wind data set (NOW-23), developed by the National Renewable Energy Laboratory (NREL), the University of Colorado, and Veer Renewables, offers the latest wind resource information for offshore regions in the United States but provides wind information for large land-based areas as well (Figure 1) (Bodini et al., 2024). The Wind Integration National Dataset (WIND) Toolkit Long-Term Ensemble Dataset (WTK-LED) Climate is a

climate-scale simulation created by Argonne National Laboratory (ANL) as part of the broader wind resource simulation package WTK-LED (Draxl et al., 2024) that is made available through NREL's web application WindWatts as annual, monthly, and diurnal wind speed averages (NREL, 2024a). All three wind datasets are freely accessible in user-friendly web applications and provide varying degrees of spatial and temporal resolution.

    As the three wind assessment datasets in this analysis are recent at the time of this manuscript, validations are limited,

particularly in coastal regions in the United States. The developers of GWA3 validated their product at 35 sites with at undefined heights in six countries (Bangladesh, Maldives, Pakistan, Papua New Guinea, Vietnam, and Zambia) and found

mean and mean absolute relative wind speed biases of -1% and 14%, respectively (DTU, 2024). At various offshore locations, Bodini et al. (2024) determined NOW-23 wind speed biases ranging from -0.42 m s$^{-1}$ to +0.11 m s$^{-1}$ for heights between 90 m and 140 m. At five land-based observational locations (one coastal and four inland) with measurement heights between 30 m and 90 m, Draxl et al. (2024) found that the ratios of WTK-LED Climate to observed wind speed probability density functions ranged from 0.86 to 0.96.

The following analysis evaluates the performance of three recent wind assessment datasets in previously unvalidated locations along United States coastlines. The validation heights (20 m – 60 m) in this work support coastal communities interested in adopting small or midsize wind energy, with turbine capacities within 1 MW (Sheridan et al., 2024). Section 2 describes the features and limitations of the three recent wind assessment datasets, along with the European Centre for Medium Range Weather Forecasts Reanalysis version 5 (ERA5) (Hersbach et al., 2020), a popular global reanalysis model widely utilised for wind energy resource assessments as a standalone product, as initial and boundary conditions for higher-resolution regional model runs, and as a reference dataset for measure-correlate-predict estimates with local observations (Olauson, 2018; Soares et al., 2020; Hayes et al., 2021; de Assis Tavares et al., 2022). ERA5 will serve as a well-validated reference model for comparing the three more recently developed wind datasets, GWA3, NOW-23, and WTK-LED Climate, all of which utilise ERA5 as the reanalysis forcing. The coastal observations are also characterized in Section 2, along with a discussion on the measure of validation used in the work. Section 3 provides the validation results of the recent wind datasets on multiple timescales, including annual average, inter-annual, seasonal, and diurnal. Section 4 summarizes the successes and challenges of each wind dataset along with recommendations for employing each in a wind resource assessment.

## 2 Data and methodology

Three recent wind datasets (GWA3, NOW-23, and WTK-LED Climate) and one reference reanalysis model (ERA5) are validated using multiple years of observations from meteorological towers along U.S. coastlines. The simulated wind data are horizontally adjusted to the observational locations using inverse distance weighting with the surrounding dataset grid points. From there, the simulated data are vertically adjusted to the observation heights using 1) the wind data output heights and temporal resolutions common to all datasets and 2) the surrounding output heights and highest temporal resolutions available per dataset. These two approaches allow us to compare the datasets both analogously and to their highest potential. The characteristics of the wind datasets and observations follow, along with a more detailed discussion of the validation process.

### 2.1 Wind assessment products

Of the three recent wind assessment products, GWA3 was developed the earliest, with its initial version released in 2015 and the third version, GWA3, in 2019 (Davis et al., 2023). The Weather Research and Forecasting (WRF) mesoscale model (Skamarock et al., 2008) was utilised with the Rapid Radiative Transfer Model (RRTM) for the longwave and shortwave

radiation schemes (Mlawer et al., 1997; Iacono et al., 2008), the Mellor-Yamada-Janjić planetary boundary layer (PBL) scheme (Janjić 1994), and ERA5 as the input and boundary conditions to produce simulated wind data at a horizontal resolution of 3 km (Davis et al., 2023). Each WRF simulation was 24 hours long with a spin-up period of six hours (Davis et al., 2023). The microscale modelling was performed using the Wind Atlas Analysis and Application Program (WAsP) model (Troen and Petersen, 1989) with an output grid spacing of 250 m, the highest horizontal spatial resolution of the recent datasets. GWA3 provides global coverage for land-based wind estimates along with offshore wind estimates within 200 km of shorelines. Wind data is output at five heights between 10 m and 200 m at annual, monthly, and diurnal temporal resolutions (Table 1). Additionally, GWA3 provides Generalized Wind Climate files that include the wind speed and wind direction distributions for a number of roughness classes that a user can incorporate into WAsP. Users can access GWA3 through its web application (DTU, 2024).

NOW-23 was published in 2023 to support offshore wind energy researchers. The dataset consists of eight regional WRF numerical simulations (three Atlantic, two Pacific, one Great Lakes, one Gulf of Mexico, and one Hawaii) (Figure 1) for which the model parameters were customized to account for geographically unique wind resource phenomena (Bodini et al., 2024). Like GWA3, NOW-23 employs RRTM for the radiation schemes and ERA5 for the reanalysis forcing. The simulations were run in 1-month segments with a spin-up period of two days (Bodini et al., 2024). The PBL scheme varies according to region between the Mellor-Yamada-Nakanishi-Niino (MYNN) (Nakanishi and Niino, 2009) and Yonsei University (YSU) (Hong et al., 2006) schemes. For the NOW-23 simulations considered in this work, MYNN is utilised for the North Pacific, Great Lakes, North Atlantic, and Mid Atlantic while YSU is utilised for the South Pacific, Gulf of Mexico, and South Atlantic (Bodini et al., 2024). NOW-23 outputs wind data at a 5-min temporal resolution and a 2-km horizontal spatial resolution at 18 heights between 10 m and 500 m (Table 1). Wind speeds output at the height of 10 m are diagnostic, derived using Monin-Obukhov similarity theory. For this analysis, NOW-23 is sampled at the top of the hour. Wind analysts can access NOW-23 and other wind datasets through the Wind Resource Database (NREL, 2024b).

WTK-LED Climate was released in 2024 as the wind climatology component of the WIND Toolkit Long-term Ensemble Dataset. The dataset uses an accelerated version of RRTM for general circulation models (RRTMG) for the radiation schemes, ERA5 for the initial and boundary conditions, and YSU for the PBL scheme (Draxl et al., 2024). WTK-LED Climate covers North America at a 4-km horizontal spatial resolution and 1-hr temporal resolution. The 20-year-long WTK-LED Climate simulations were reinitialized each year and run for a total of 14 months continuously without nudging with the last two months from previous years (November and December) as spin-up time (Draxl et al., 2024). Through WindWatts (NREL, 2024a), users can access WTK-LED Climate annual, monthly, and diurnal wind speed data at seven output heights between 30 m and 140 m (Table 1). The WTK-LED Climate wind data at 10 m was provided by the dataset developers to support this analysis.

ERA5 is a widely used global reanalysis model (Hersbach et al., 2020) in the wind energy community that began initial production in 2016. The single level ERA5 product outputs wind data at 10 m and 100 m above ground level (Table 1). The winds at the 10 m level are obtained via interpolation between the lowest model level and the surface and are corrected to

align with open terrain observations. To adjust to the observations, the correction procedure for the ERA5 10 m winds involves an aerodynamic roughness length that is typical for open terrain with grassland (ECMWF, 2016).

**Table 1.** Characteristics of the wind assessment datasets analysed along U.S. coastlines.

| Dataset | GWA3 | NOW-23 | WTK-LED Climate[b] | ERA5 |
|---|---|---|---|---|
| **Developers** | DTU, World Bank Group | NREL, University of Colorado, Veer Renewables | ANL | ECMWF |
| **Temporal Coverage** | 2008 - 2017 | 2000 – 2019[a] | 2001 – 2020 | 1940 – present |
| **Temporal Resolution** | Annual average wind speeds and normalized wind speed indices for establishing wind speed trends according to hour of day, month of year, and specific year in the 10-year coverage period | 5-min | Average wind speed by month and hour of day (12 x 24) for each year in the 20-year coverage period | 1-hr |
| **Spatial Coverage** | Global | U.S. marine regions | North America | Global |
| **Spatial Resolution** | 0.25-km | 2-km | 4-km | 0.25°[c] |
| **Wind Output Heights** | 10 m, 50 m, 100 m, 150 m, 200 m | 10 m, 20 m, 40 m, 60 m, 80 m, 100 m, 120 m, 140 m, 160 m, 180 m, 200 m, 220 m, 240 m, 260 m, 280 m, 300 m, 400 m, 500 m | 30 m, 40 m, 60 m, 80 m, 100 m, 120 m, 140 m | 10 m, 100 m for the single levels product |

[a] Depending on the region, the temporal extent varies between 2019 and 2022. For all regions analysed in this work, the temporal extent of NOW-23 is 2019.

[b] WTK-LED Climate characteristics in this work reflect the data available via WindWatts (NREL, 2024a) at the time of article submission, with the addition of the WTK-LED Climate data at the 10 m output height, which was provided to researchers to support this analysis.

[c] The ERA5 data have been converted from the native reduced Gaussian grid to a regular latitude-longitude grid at 0.25° (Hersbach et al., 2020).

## 2.2 Wind observations

To evaluate the recent wind assessment datasets, 23 wind speed observational datasets from meteorological towers with measurements at heights relevant to small and midsize distributed wind turbine hub heights are used. The observations are sourced from the National Data Buoy Center (19 sites), the Bonneville Power Administration (3 sites), and a collaborative

project between the U.S. Department of Energy and a wind energy industry partner (1 site) (Figure 1), and all but the last observational dataset are publicly available as outlined in the data availability section. A comprehensive guide to the observational sites is provided in Appendix A.

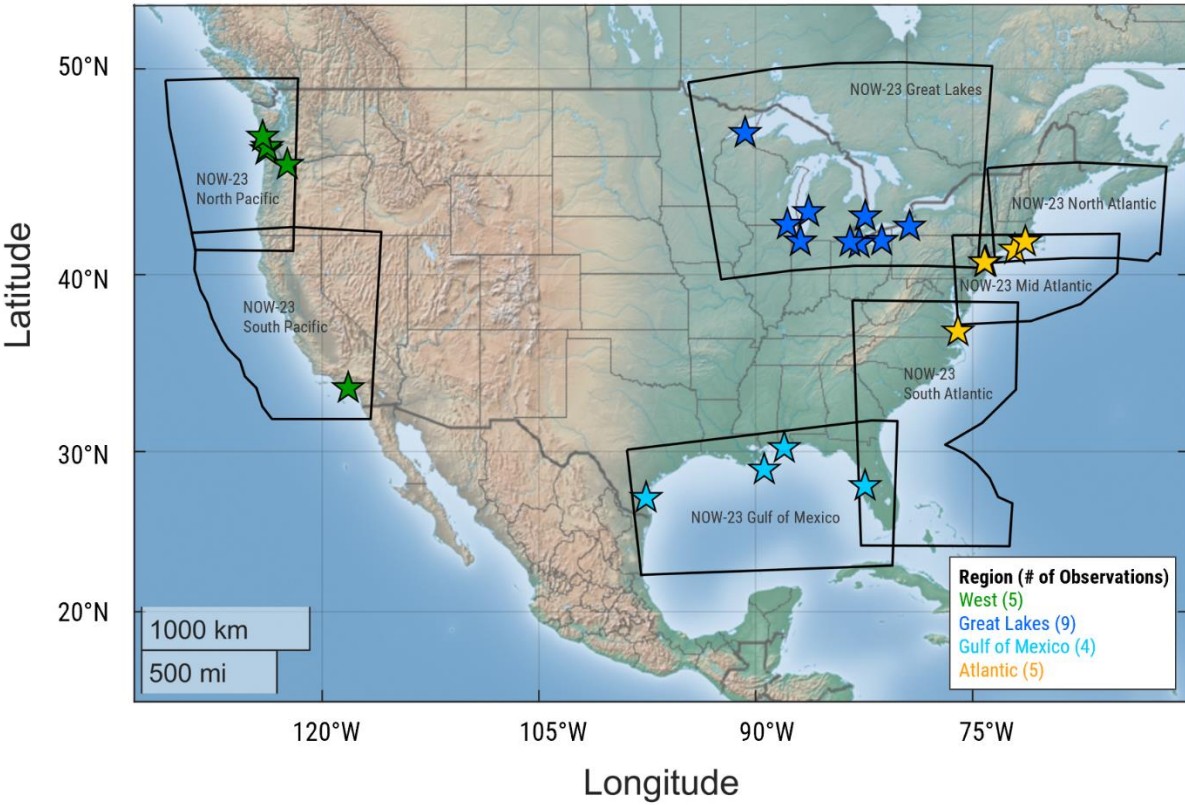

**Figure 1.** Map of observational locations and regional designations used in this analysis, along with NOW-23 domain definitions.

For analysis according to region, the observations are grouped as follows: West (5 sites), Great Lakes (9 sites), Gulf of Mexico (4 sites), and Atlantic (5 sites) (Figure 1). While additional observations are available in these regions, they are not included in this study due to 1) data quality concerns (criteria in following paragraph), 2) minimal or non-existent overlap between observation and dataset temporal coverage periods, or 3) measurement heights falling outside the bounds of this analysis. In this work, measurement heights span the hub heights of small and midsize distributed wind turbines, from 20 m

to 60 m above ground level, with the majority occurring between 20 m and 30 m (77%) (Figure 2a). Annual average wind speeds at the observational sites range from 2.5 m s$^{-1}$ to 7.0 m s$^{-1}$ (Figure 2b). The observational sites span 16 states and are located within four kilometres of a body of water, including the Atlantic and Pacific Oceans, the Gulf of Mexico, the Columbia River, and four of the five Great Lakes (Erie, Huron, Michigan, and Superior) (Figure 1, Figure 2c). When considering the distribution of flow direction within a 100 km radius to represent the extent of onshore and offshore breezes

(Gille et al., 2005; Viner et al., 2021), the winds at 14 sites predominantly originate over land while the winds at 9 sites predominantly originate over water (Figure 2e, Appendix A) as determined by the Global Land Cover and Land Use Change 2000-2020 (Potapov et al., 2022) and the wind roses for each site.

   To establish robust validation reference datasets, the wind speed observations are subject to quality control by removing instances or periods of atypical or unphysical reported wind speeds (less than 0 m s$^{-1}$, greater than 50 m s$^{-1}$, or non-varying

periods of time greater than 4 hours) that could be indicative of instrument error due to an outage or weather impacts like icing. In order to temporally align with the wind assessment datasets (Section 2.1), only observations between the years 2008 and 2017 are utilised in this study. Because GWA3 outputs wind speed information on an annual resolution, the wind observations need to be representative of full calendar years. Therefore, only calendar years with 95% or greater of observational data recovery and quality are retained for the comparison (Figure 2d).


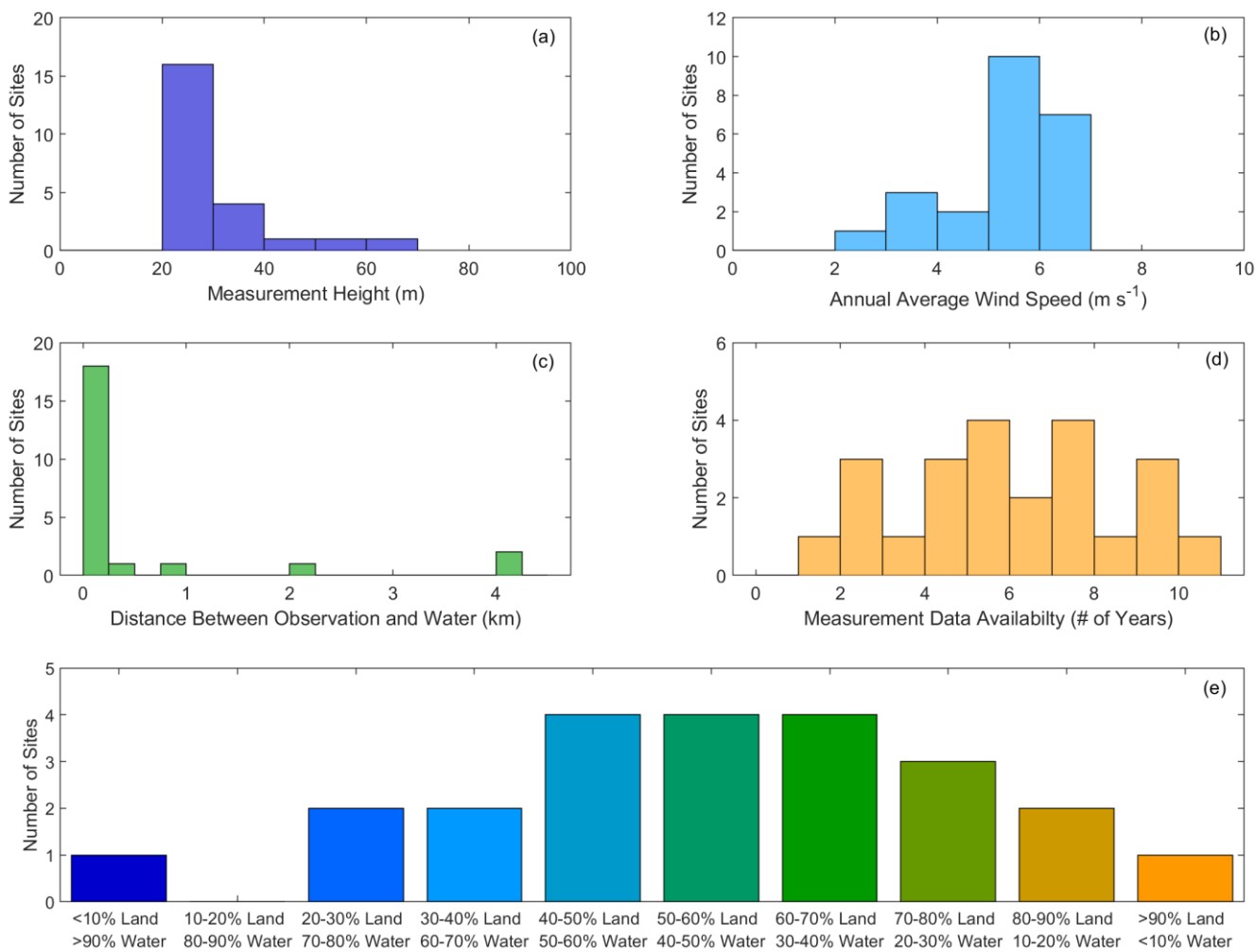

**Figure 2.** (a) Measurement height, (b) annual average wind speed across the measurements, (c) distance between observational location and the nearest water body, (d) length of measurement data available, and (e) distribution of wind originating from land and water within 100 km of the observational location across 23 coastal sites.

## 2.3 Validation methodology

Prior to validating the recent and reference datasets at the coastal locations in this work, they must be adjusted to the observational characteristics. Horizontally, each dataset is adjusted to the location of the coastal meteorological tower by using inverse distance weighting. Inverse distance weighting is selected because the frequently dense wind speed contours along coastlines (Sheridan et al., 2022a) eliminate using the nearest-neighbour grid point as a representative baseline for comparing simulated and observed wind speeds, particularly for coarser datasets. Vertically, each dataset is adjusted to the observational heights via the power law using the dataset wind speeds ($u_{lo}$, $u_{hi}$) at surrounding output heights ($z_{lo}$, $z_{hi}$) to the observation height ($z_{obs}$). This method is selected based on the study of Duplyakin et al. (2021), who found that the power law minimized errors due to vertical adjustment of wind dataset output heights to observation heights. Throughout the study, the surrounding output heights and temporal frequency of calculation of the shear exponent $\alpha$ (Eq. 1) and the adjusted dataset wind speed $u_{mod}$ (Eq. 2) will be considered according to both the common dataset characteristics (i.e., annual average wind speeds at the 10 m and 100 m output heights) and to each dataset's highest resolution potential to implicitly account for how $\alpha$ varies with atmospheric stability.

$$\alpha = \left. \ln(u_{hi}/u_{lo}) \middle/ \ln(z_{hi}/z_{lo}) \right. \tag{1}$$

$$u_{mod} = u_{hi} \left( z_{obs}/z_{hi} \right)^{\alpha} \tag{2}$$

Temporally, the datasets and observations are aligned according to time period and averaging period. For example, since GWA3 and WTK-LED Climate output annual, monthly, and hourly-average wind data (Table 1), the observations and higher-resolution datasets (NOW-23 and ERA5) are similarly adjusted to annual, monthly, and hourly averages prior to comparison with GWA3 and WTK-LED Climate (Table 1).

Three key error metrics are utilised in this study to assess the performance of the recent wind datasets in a manner that is useful for a coastal community or resident wishing to assess their wind energy potential. First, the wind speed bias informs on whether each dataset tends to overestimate (positive bias) or underestimate (negative bias) the observed wind resource over a period of time $N$ (Eq. 3). The relative error is the absolute difference between the simulated and observed wind speeds normalized by the observed wind speed, providing detail on the magnitude of error in each dataset (Eq. 4). Finally, to characterize the accuracy of the datasets according to temporal trends such as inter-annual, seasonal, and diurnal, the Pearson correlation coefficient explains the degree to which the simulated and observed wind speeds are linearly related (Eq. 5).

$$bias = \frac{1}{N} \sum_{i=1}^{N} \left( u_{mod,i} - u_{obs,i} \right) \tag{3}$$

$$relative\ error = 100\% * \frac{|\overline{u_{mod}} - \overline{u_{obs}}|}{\overline{u_{obs}}} \tag{4}$$

$$correlation = \frac{\sum_{i=1}^{N} \left( u_{mod,i} - \overline{u_{mod}} \right)\left( u_{obs,i} - \overline{u_{obs}} \right)}{\sqrt{\sum_{i=1}^{N} \left( u_{mod,i} - \overline{u_{mod}} \right)^2} \sqrt{\sum_{i=1}^{N} \left( u_{obs,i} - \overline{u_{obs}} \right)^2}} \tag{5}$$

## 3 Results

The following sections compare recent wind assessment dataset performance at coastal sites versus the more established ERA5 in order to enable dataset users with the level of accuracy they can expect in representation of important pre-construction wind metrics, such as annual average wind speed and temporal trends in the wind resource. To start, the datasets are validated analogously by adjusting the annual average dataset output wind speeds at the common heights (10 m and 100 m) to the observational heights (Table 2). This initial analysis is intended to ensure consistency across the datasets in the

temporal and vertical spaces to evaluate the performance of the datasets, with their differences in horizontal spatial resolution, in geographically complex coastal environments. The study progresses to assess the impacts of increasing temporal and vertical spatial output resolution on dataset performance (Table 2). Later, each dataset's performance is evaluated at the regional, seasonal, diurnal, and inter-annual levels.

**Table 2.** Scenarios for determining the shear exponent for adjusting simulated wind speeds at dataset output heights to observational heights.

| Scenario | Description | GWA3 | NOW-23 | WTK-LED Climate | ERA5 |
|---|---|---|---|---|---|
| 1 | Analogous calculation using annual average wind speeds at output heights shared by all datasets (10 m and 100 m) | For each year, $\alpha = \dfrac{\ln(\overline{u_{100m}}/\overline{u_{10m}})}{\ln(100/10)}$ | For each year, $\alpha = \dfrac{\ln(\overline{u_{100m}}/\overline{u_{10m}})}{\ln(100/10)}$ | For each year, $\alpha = \dfrac{\ln(\overline{u_{100m}}/\overline{u_{10m}})}{\ln(100/10)}$ | For each year, $\alpha = \dfrac{\ln(\overline{u_{100m}}/\overline{u_{10m}})}{\ln(100/10)}$ |
| 2 | Calculation using annual average wind speeds at the nearest surrounding heights to each observation | For each year, $\alpha = \dfrac{\ln(\overline{u_{hi}}/\overline{u_{lo}})}{\ln(z_{hi}/z_{lo})}$ $z_{lo} = 10$ or $50$ m $z_{hi} = 50$ or $100$ m according to $z_{obs}$ | For each year, $\alpha = \dfrac{\ln(\overline{u_{hi}}/\overline{u_{lo}})}{\ln(z_{hi}/z_{lo})}$ $z_{lo} = 10, 20,$ or $40$ m $z_{hi} = 20, 40,$ or $60$ m according to $z_{obs}$ | For each year, $\alpha = \dfrac{\ln(\overline{u_{hi}}/\overline{u_{lo}})}{\ln(z_{hi}/z_{lo})}$ $z_{lo} = 10, 30,$ or $40$ m $z_{hi} = 30, 40,$ or $60$ m according to $z_{obs}$ | For each year, $\alpha = \dfrac{\ln(\overline{u_{hi}}/\overline{u_{lo}})}{\ln(z_{hi}/z_{lo})}$ $z_{lo} = 10$ m, $z_{hi} = 100$ m for all observations |
| 3 | Calculation at each dataset's highest temporal resolution using the nearest surrounding heights to each observation | For each year, $\alpha = \dfrac{\ln(\overline{u_{hi}}/\overline{u_{lo}})}{\ln(z_{hi}/z_{lo})}$ $z_{lo} = 10$ or $50$ m $z_{hi} = 50$ or $100$ m according to $z_{obs}$ | At each hour, $\alpha = \dfrac{\ln(u_{hi}/u_{lo})}{\ln(z_{hi}/z_{lo})}$ $z_{lo} = 10, 20,$ or $40$ m $z_{hi} = 20, 40,$ or $60$ m according to $z_{obs}$ | At each month/hour combination, $\alpha = \dfrac{\ln(\overline{u_{hi}}/\overline{u_{lo}})}{\ln(z_{hi}/z_{lo})}$ $z_{lo} = 10, 30,$ or $40$ m $z_{hi} = 30, 40,$ or $60$ m according to $z_{obs}$ | At each hour, $\alpha = \dfrac{\ln(u_{hi}/u_{lo})}{\ln(z_{hi}/z_{lo})}$ $z_{lo} = 10$ m, $z_{hi} = 100$ m for all observations |

### 3.1 Annual average wind speed

Using the common model output heights of 10 m and 100 m to establish the simulated wind profiles for alignment to the

coastal observational heights, ERA5 (lowest horizontal spatial resolution) and GWA3 (highest horizontal spatial resolution)

have the lowest annual average wind speed biases with medians of -0.23 m s$^{-1}$ and +0.54 m s$^{-1}$, respectively, while NOW-23 and WTK-LED Climate have similar medians of 0.85 m s$^{-1}$ and 0.84 m s$^{-1}$ (Figure 3a). GWA3, NOW-23, and WTK-LED Climate predominantly overestimate the observed annual average wind speeds (overestimating at 78%, 78%, and 96% of the coastal sites, respectively), whereas ERA5 follows a well-documented trend of underestimating the wind resource (Ramon et al., 2019; Murcia et al., 2022; Sheridan et al., 2022b; Wilczak et al., 2024) at 61% of the sites in this analysis. In terms of relative error, GWA3 is the best performing dataset with a median of 9.9%, followed by ERA5 (10.0%), NOW-23 (14.1%), and WTK-LED Climate (18.5%) (Figure 3b).

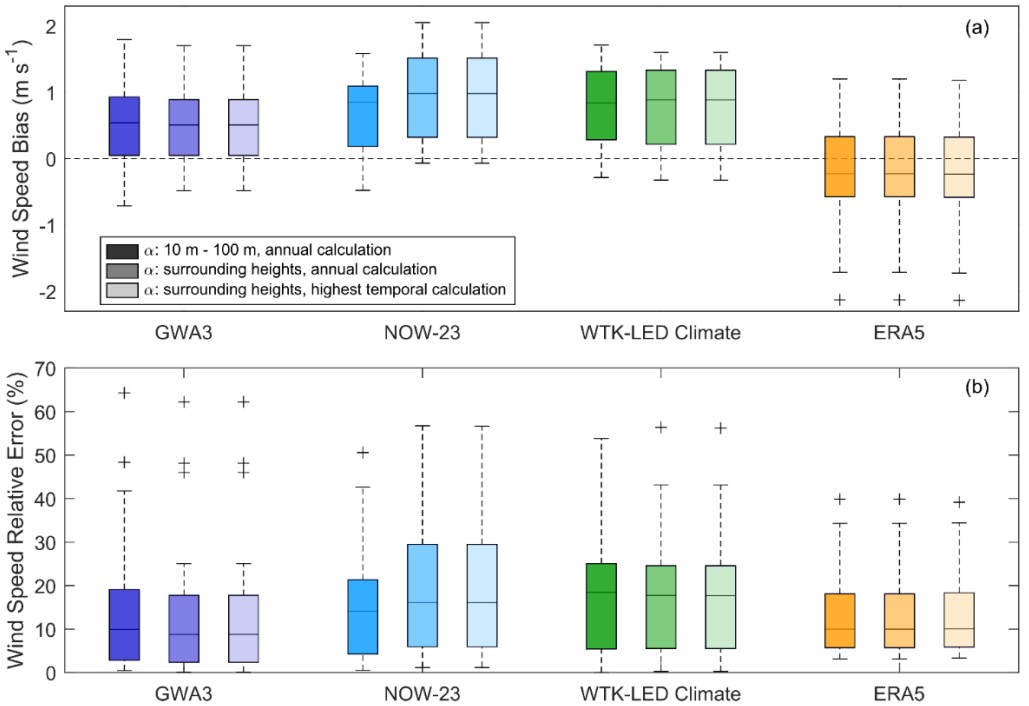

**Figure 3.** Annual average wind speed (a) biases and (b) relative errors across 23 coastal observational locations, with adjustment to observational height performed 1) annually using the common dataset output heights of 10 m and 100 m, 2) annually using the nearest surrounding output heights to the observation height, and 3) at each dataset's highest temporal resolution using the nearest surrounding output heights to the observation height.

While the above study is interesting from a scientifically comparative standpoint, it is anticipated that a wind assessment dataset user will desire to take advantage of the highest vertical spatial and temporal resolutions available in the datasets. Beginning with the vertical, Figure 3 also presents the annual average wind speed biases and relative errors resulting from adjusting the dataset output to the observational heights using the nearest surrounding dataset output heights (Table 1). While the GWA3 and WTK-LED Climate relative errors improve on using the surrounding dataset output heights versus 10 m and 100 m, with median relative errors of 8.8% versus 9.9% for GWA3 and 17.8% versus 18.5% for WTK-LED Climate, the NOW-23 relative errors noticeably degrade upon using the surrounding dataset output heights (median relative error of

16.2% versus 14.1% using 10 m and 100 m). Narrowing the dataset adjustment height range from 10 m and 100 m to the surrounding levels increases the number of sites where NOW-23 overestimates the annual average wind speed from 78% to 96%.

Our first inclination to try and understand the notable differences in NOW-23 performance according to output heights for adjustment to observational height was to examine the wind shear between the surrounding dataset output heights. Following

the trend of consistent performance noted for GWA3 and WTK-LED Climate, the shear exponents for these datasets using the surrounding heights differ little from their counterparts using 10 m and 100 m (median differences of 0.01 and 0.02, respectively). The NOW-23 wind shear exponents differ more substantially using the nearest surrounding heights to the observation heights versus 10 m and 100 m (median difference of 0.05), however, not in the anticipated direction. Given the increase in wind speed overestimation noted when using the surrounding heights for adjustment, it was initially suspected

that NOW-23 was potentially overestimating the wind shear between the surrounding heights; however, the opposite result of reduction in the wind shear exponents was determined between the surrounding NOW-23 heights as compared to 10 m and 100 m. This result prompted a more expansive look at the NOW-23 wind profiles.

The relatively large NOW-23 wind shear is not found between the output heights surrounding the observational heights (20 m, 40 m, and 60 m for the observational collection utilised in this study), but rather between the two lowest NOW-23

output heights (10 m and 20 m) for many of the observational locations. Figure 4 shows that the shear exponents calculated between the lowest output heights and between 10 m and 100 m are quite similar for GWA3 and WTK-LED Climate (median differences within 0.02), whereas shear exponents using the lowest NOW-23 output heights tend to be larger than their counterparts calculated between 10 m and 100 m by a median of 0.10. The relatively large wind shear between the two lowest NOW-23 output heights (10 m and 20 m) corresponds with larger NOW-23 wind speeds at 20 m, which is the lower

surrounding output height for 87% of the observations in this analysis. Annual average wind speed profiles based on all available dataset output heights and just on 10 m and 100 m from GWA3, NOW-23, and WTK-LED Climate are provided in Figure 5 for three distinct geographic observational locations (Washington, Indiana, and Connecticut) to illustrate the impacts of the large wind shear between the lowest NOW-23 output heights.

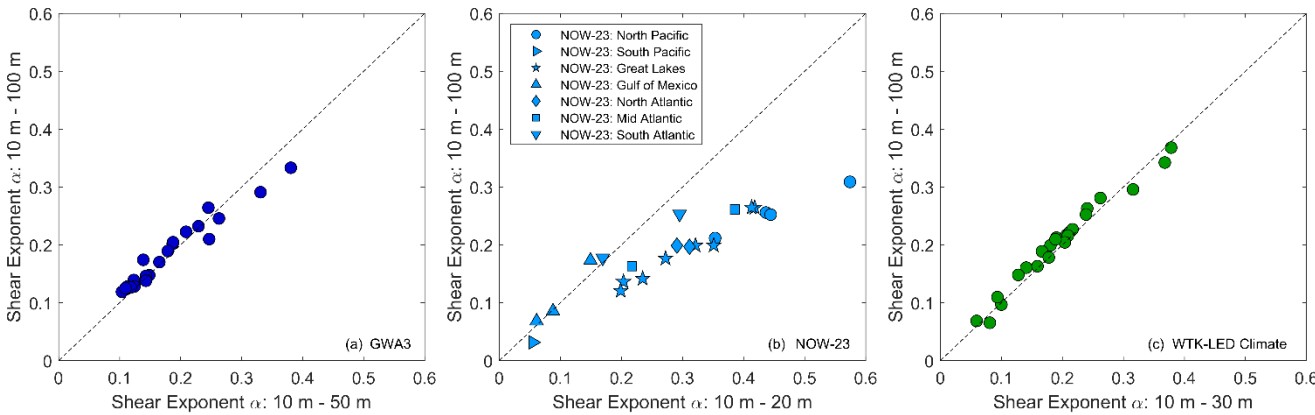

**Figure 4.** Shear exponents based on the lowest dataset output heights (x-axis) and 10 m and 100 m (y-axis) across 23 coastal sites from (a) GWA3, (b) NOW-23, and (c) WTK-LED Climate.

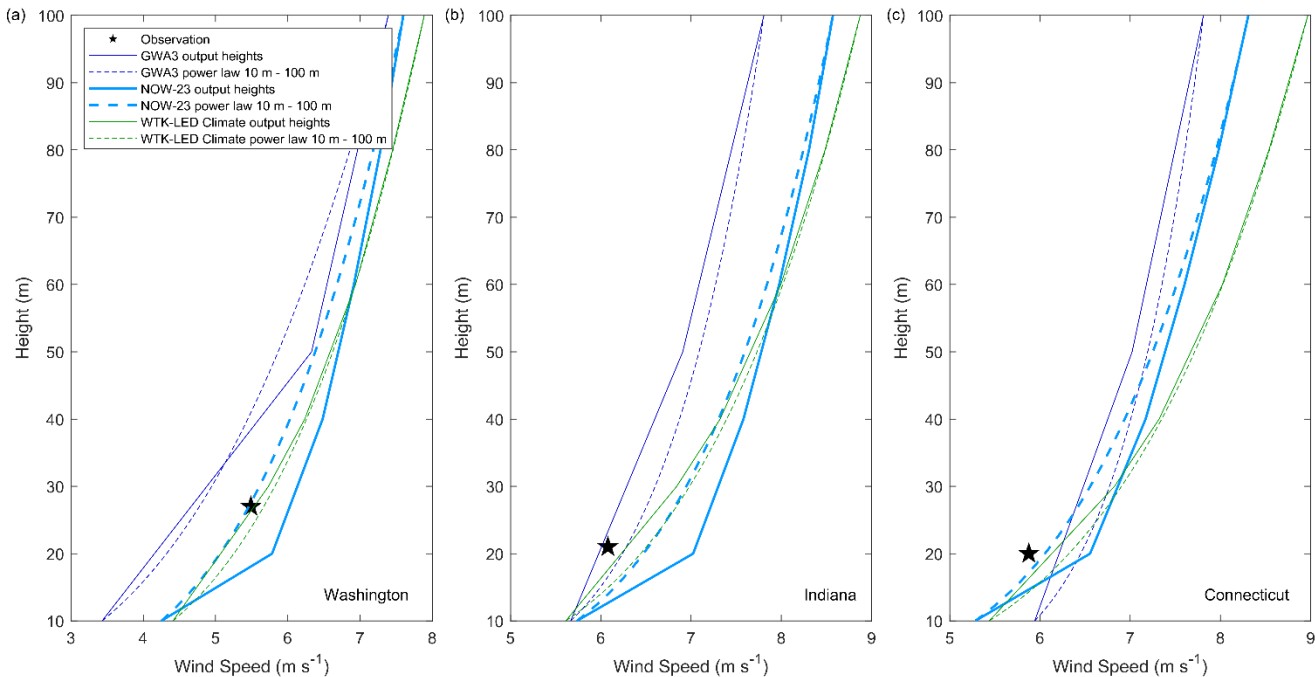

**Figure 5.** Simulated wind speed profiles at observational locations in (a) Washington, (b) Indiana, and (c) Connecticut.

When each dataset is adjusted to the observation height using its full temporal and vertical spatial capabilities, i.e., the shear exponent is calculated at the highest available temporal frequency using the nearest surrounding output heights, the relative errors change minimally from those calculated using the lowest temporal frequency, with medians of 8.8% (GWA3), 10.0% (ERA5), 16.1% (NOW-23), and 17.7% (WTK-LED Climate). The recent datasets tend to overestimate the wind resource (GWA3 at 78% of the observation sites and NOW-23 and WTK-LED Climate at 96% of the observation sites).

Disagreement between actual and predicted wind energy generation can lead to customer dissatisfaction and damage to the reputation of distributed wind as a viable energy resource, particularly in circumstances of overestimation. The findings of this work encourage users of GWA3, NOW-23, and WTK-LED Climate for coastal analyses to adjust their annual average wind speed and wind energy production expectations. Additionally, the findings encourage the use of bias correction where possible, which can provide significant improvement to wind resource estimates (Wilczak et al., 2024). The analyses throughout the remainder of the manuscript evaluate the datasets according to region and temporal trends in the wind resource at the coastal observation sites using the height adjustment with full temporal and spatial capabilities.

The dataset with the greatest horizontal spatial resolution, GWA3 at 250 m (Table 1), provided the smallest relative annual average wind speed errors (Figure 3). Therefore, it is of interest to investigate to the degree possible whether GWA3's performance can be attributed to improved representation of the land and water boundaries at the complex coastal observation heights or simply reduced bias compared with the other recent wind datasets. As a simple exercise, the annual average wind speed relative errors using GWA3 were established using horizontal sampling resolutions of 250 m, 2 km (the horizontal resolution of NOW-23), 4 km (the horizontal resolution of WTK-LED Climate), and 30 km (approximately the horizontal resolution of ERA5 in the continental United States). The wind speed relative errors from GWA3 at the coastal locations vary little for horizontal resolutions within 4 km, and not following the expected trend of lower errors with higher resolution (medians = 8.8% at 250 m, 8.2% at 2 km, and 5.8% at 4 km). Even at a horizontal resolution of 30 km, GWA3 produces a median wind speed relative error of 8.3%. This result is likely due to the GWA3 flow being more accurately modelled using a high-resolution grid, so that even with the removal of some grid points, the solution is still good.

While no trends emerged for GWA3 performance according to horizontal sampling resolution, the recent wind datasets show distinct differences in representing the annual average wind speed relative errors according to whether the coastal sites are dominated by land or water. Of the 23 coastal sites in this analysis, 14 have wind flow distributions where most of the wind is arriving from land, while 9 have wind flow distributions where most of the wind is arriving from water (Figure 2e). Each region (Figure 1) is represented in both the water-dominant and land-dominant lists of sites according to flow. GWA3, NOW-23, WTK-LED Climate, and ERA5 perform notably better for the sites with water-dominant wind distributions, with median wind speed relative errors of 3.1%, 9.3%, 15.0%, and 8.6% respectively, than for the sites with land-dominant wind distributions, where the median relative errors are 15.8%, 25.5%, 18.3%, and 12.0% (Figure 6b). The significant decrease in dataset accuracy for land-dominant sites is likely due to a combination of challenges, including dataset representation of complex terrain (particularly for the western sites) and characterization of surface roughness length. Concerning the latter, the land-dominant sites tend to have wind flow distributions that favour cropland, forests, and built environments (Table A.1) which have greater roughness lengths than, for example, the open terrain grassland roughness length utilised for post-processing ERA5's 10 m single level output (ECMWF, 2016). All datasets follow the same trend of increasingly positive wind speed biases for land-dominant sites relative to the water-dominant sites. For GWA3, NOW-23, and WTK-LED Climate, the sites with land-dominant wind distributions experience a greater degree of dataset overestimation, with median wind speed biases of 0.81 m s$^{-1}$, 1.33 m s$^{-1}$, and 0.98 m s$^{-1}$, respectively, while the median wind speed biases for the water-

dominant sites are 0.05 m s$^{-1}$, 0.36 m s$^{-1}$, and 0.71 m s$^{-1}$ (Figure 6a). For ERA5, the degree of model underestimation is reduced for the land-dominant sites relative to the water-dominant sites, with median wind speed biases of -0.19 m s$^{-1}$ and -0.54 m s$^{-1}$.

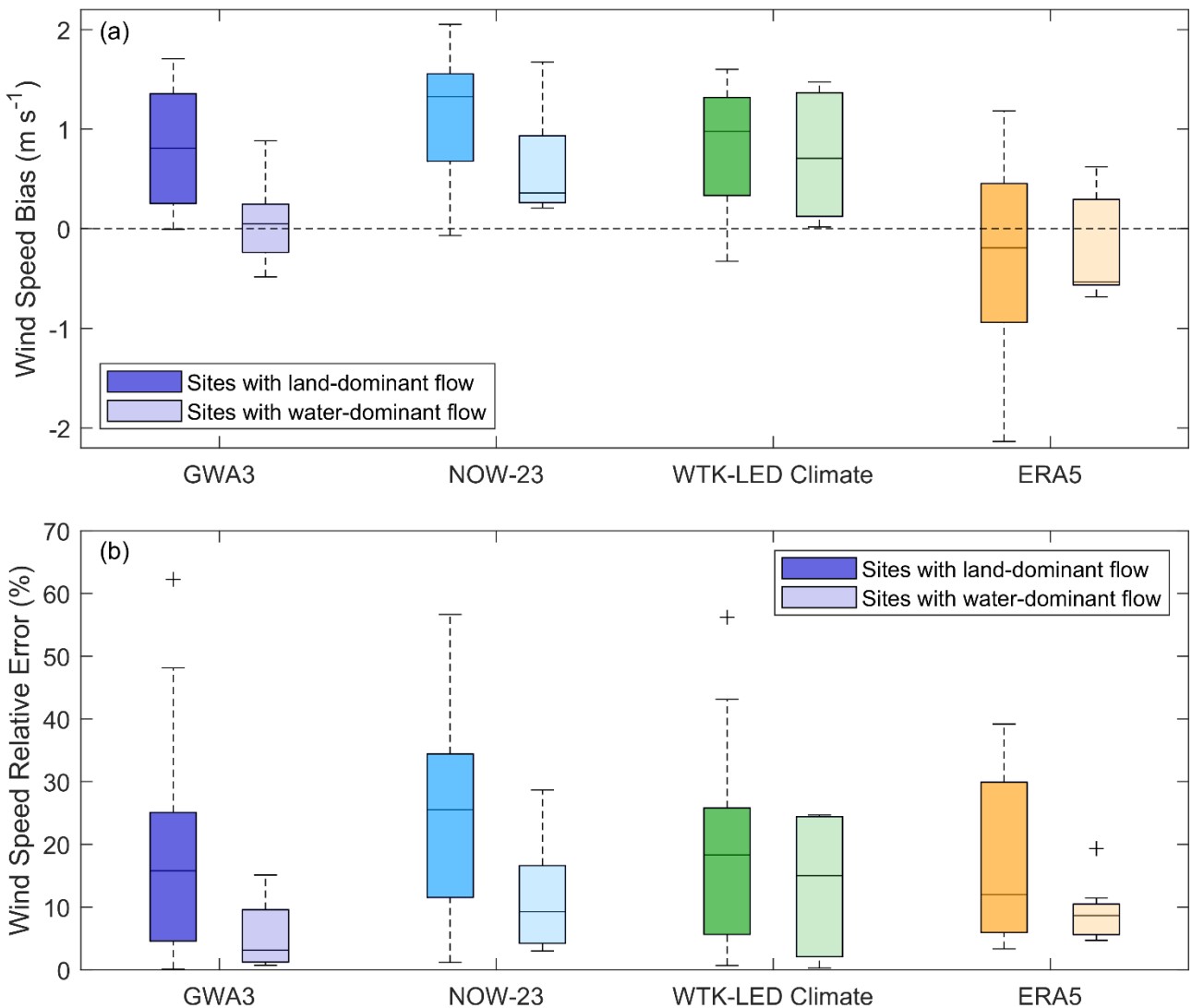

**Figure 6.** Annual average wind speed (a) biases and (b) relative errors across the 14 coastal sites with land-dominant wind flow and the 9 coastal sites with water-dominant wind flow.

### 3.2 Regional performance

Across the regions, ERA5 consistently produces the smallest biases, with trends of underestimating the observed wind resource in the West and Atlantic regions, overestimating at sites along the Great Lakes, and exhibiting little bias in the Gulf

of Mexico (medians = -0.55 m s$^{-1}$, -0.38 m s$^{-1}$, 0.28 m s$^{-1}$, and 0.01 m s$^{-1}$, respectively) (Figure 7). With the exception of the Gulf of Mexico, where GWA3 performs similarly to ERA5 (median bias = 0.01 m s$^{-1}$), the three recent wind datasets overestimate the observed annual average wind speeds in all regions, with significant overestimation noted in most regions (using a criteria of 0.5 m s$^{-1}$). The strongest recent dataset overestimation is found for sites along the Great Lakes, where the median biases range from 0.82 m s$^{-1}$ (GWA3) to 1.55 m s$^{-1}$ (NOW-23) (Figure 7). Interestingly, Bodini et al. (2024) determined that NOW-23 underestimated the observed wind speed by 0.42 m s$^{-1}$ at 105 m altitude at an offshore location in Lake Michigan.

In terms of relative error, which focuses on the magnitude of error relative to the observed wind speed, dataset performance is less consistent than for the trends in bias (Figure 7). ERA5 provides the smallest relative errors in the West, followed closely by NOW-23 and GWA3, with median relative errors of 10%, 11%, and 13%, respectively. For sites along the Great Lakes, ERA5 again produces the lowest relative errors (median = 10%), whereas the recent datasets provide median relative errors ranging from 15% (GWA3) up to 29% (NOW-23). Across the four sites in the Gulf of Mexico region, GWA3 exhibits the lowest relative errors, with a median of 3%, and NOW-23 and ERA5 similarly produce median relative errors under 10% (5% and 7%, respectively). WTK-LED Climate provides the lowest relative errors (median = 7%) for sites along the Atlantic Ocean, followed closely by GWA3, ERA5, and NOW-23 (medians = 9%, 11%, and 12%).

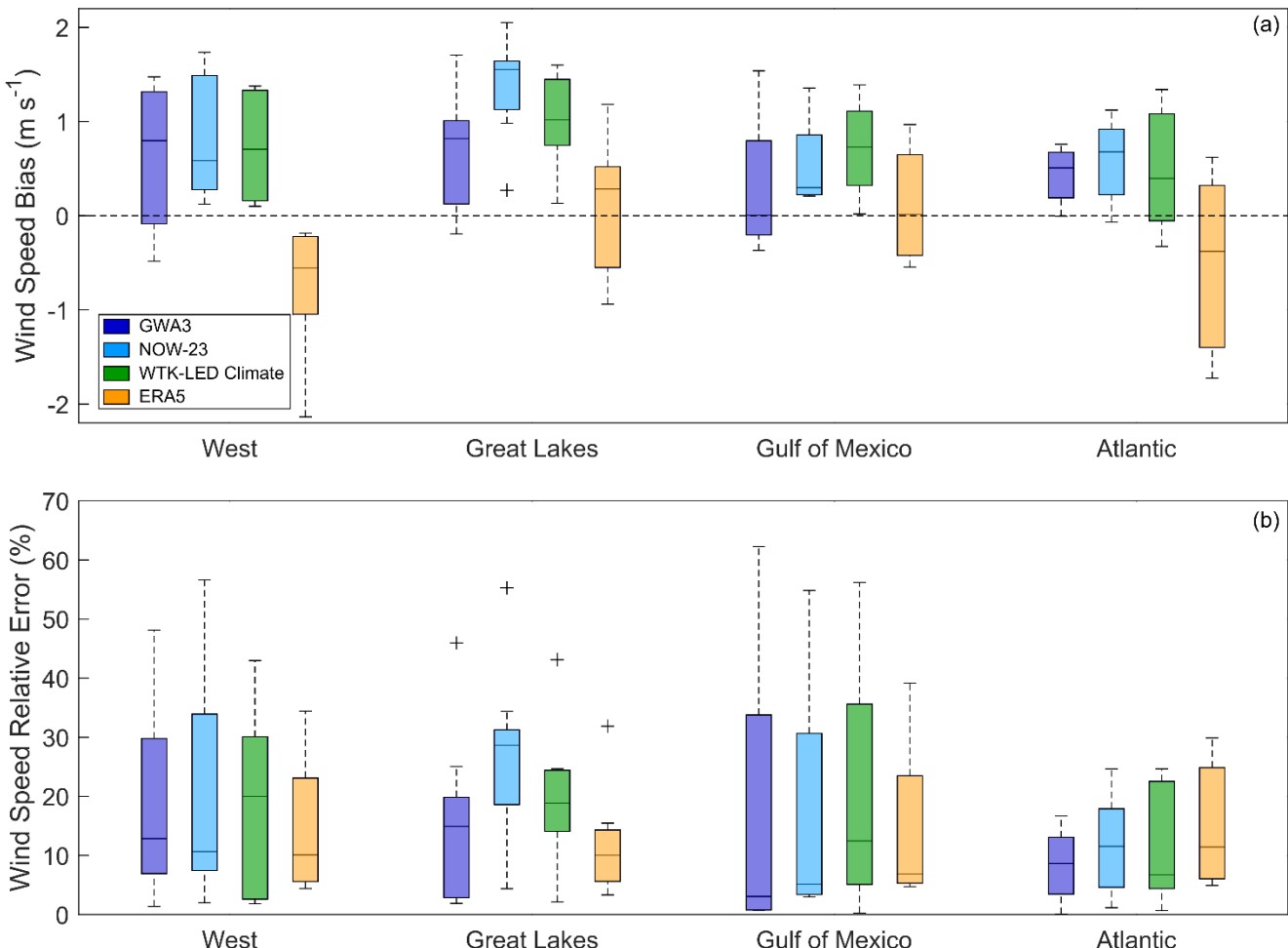

**Figure 7.** Annual average wind speed (a) biases and (b) relative errors according to geographic region.

### 3.3 Seasonal and diurnal performance

Two approaches to assessing wind dataset performance in representing the seasonal wind cycles at the coastal observational locations are presented in Figure 8. First, the observed normalized monthly wind speeds (using whichever years between 2008 and 2017 meet the data recovery and quality requirements in Section 2.2) are compared with the simulated normalized monthly wind speeds during the full decade of 2008-2017. This initial comparison allows for the analysis of GWA3's performance in seasonal wind cycle representation, as GWA3 does not provide monthly trends in the wind speed for individual years. For the second analysis, the simulated monthly wind speeds from NOW-23, WTK-LED Climate, and ERA5 are temporally aligned with those from the observations using only the years between 2008 and 2017 where the observations meet the data recovery and quality requirements.

Across the recent datasets and ERA5, NOW-23 best represents the seasonal wind cycles at the 23 coastal sites, with
correlations with observations of 0.92 using the full decade of dataset data and 0.94 using the years temporally aligned with the observations (Figure 8). ERA5 is the next most successful dataset at seasonal representation (correlations = 0.90, 0.92), followed by WTK-LED Climate (correlations = 0.89, 0.88), and GWA3 (correlation = 0.84).

The site with the most pronounced seasonal cycle (normalized monthly wind speeds ranging from 0.68 to 1.61, $\Delta \bar{u}_{monthly} = 0.93$) is located in Oregon along the Columbia River (Figure 9a). Both GWA3 and NOW-23 correctly identify this site as
having the most pronounced seasonal wind speed cycle of the 23 sites, regardless of whether a full decade of simulated data is considered or just the overlapping years with the observations. Both WTK-LED Climate and ERA5 incorrectly assign the site with the most pronounced seasonal wind speed cycle, regardless of temporal period. When considering a full decade of simulated data, WTK-LED Climate and ERA5 determine the site with the most pronounced seasonal cycle to be located in the Great Lakes (observed $\Delta \bar{u}_{monthly} = 0.67$) and Gulf of Mexico (observed $\Delta \bar{u}_{monthly} = 0.50$) regions, respectively. When
considering the overlapping observational years, WTK-LED Climate and ERA5 identify the site with the most pronounced seasonal cycle to exist in the West, though at a site along the Pacific Ocean (observed $\Delta \bar{u}_{monthly} = 0.43$).

The site with the least pronounced seasonal cycle (normalized monthly wind speeds ranging from 0.88 to 1.14, $\Delta \bar{u}_{monthly} = 0.26$) is located along a Florida inlet (Figure 9b). NOW-23 correctly identifies this site as having the least pronounced seasonal wind speed cycle of the analysed sites, regardless of whether a full decade of simulated data is considered or just
the overlapping years with the observations. WTK-LED Climate correctly identifies the site when considering the overlapping observational period and determines a site in Texas (observed $\Delta \bar{u}_{monthly} = 0.36$) when considering the full decade. GWA3 chooses a site in the Atlantic region (observed $\Delta \bar{u}_{monthly} = 0.30$) as having the least pronounced seasonal cycle, and ERA5 identifies sites in California and Rhode Island when considering the full decade and overlapping observational period (observed $\Delta \bar{u}_{monthly} = 0.37, 0.31$), respectively.

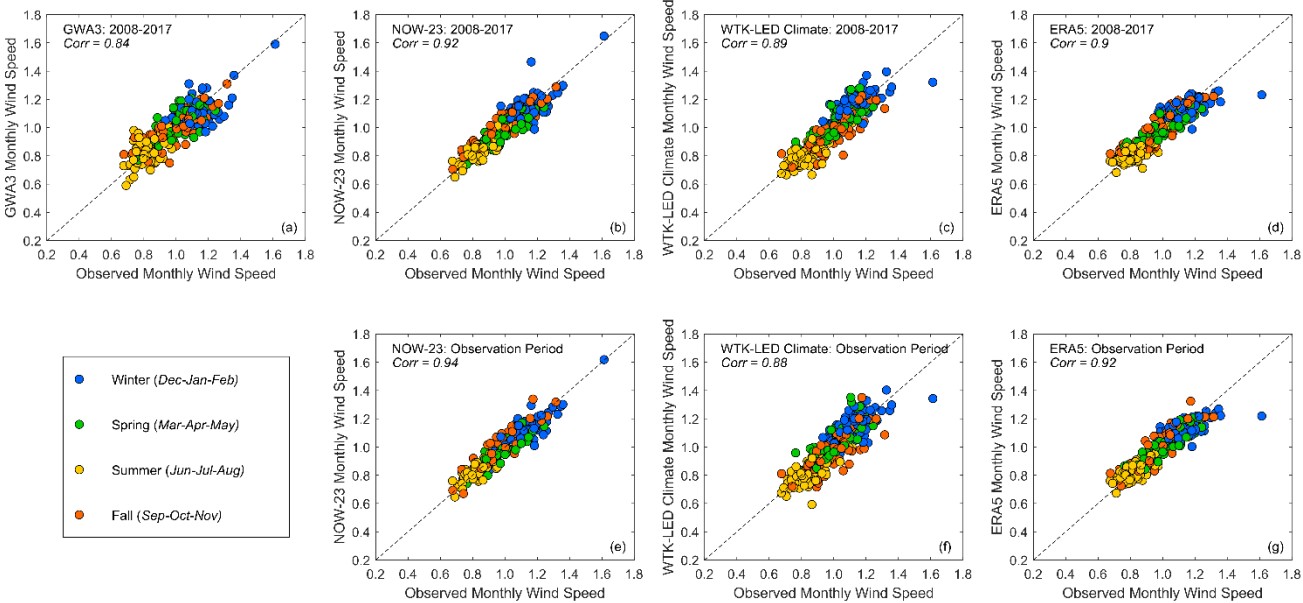

**Figure 8.** Observed and simulated normalized monthly average wind speeds (monthly average wind speed / annual average wind speed) across 23 coastal sites. Panels (a), (b), (c), and (d) depict the observed monthly average wind speeds versus the simulated monthly average wind speeds based on a decade (2008-2017) of dataset data from GWA3, NOW-23, WTK-LED Climate, and ERA5, respectively. This analysis is included to allow for comparison with GWA3, which does not provide monthly trends in the wind speed for individual years. Panels (d), (e), and (f) share the observed monthly average wind speeds versus the simulated monthly average wind speeds calculated during the temporal period of the observations from NOW-23, WTK-LED Climate, and ERA5.

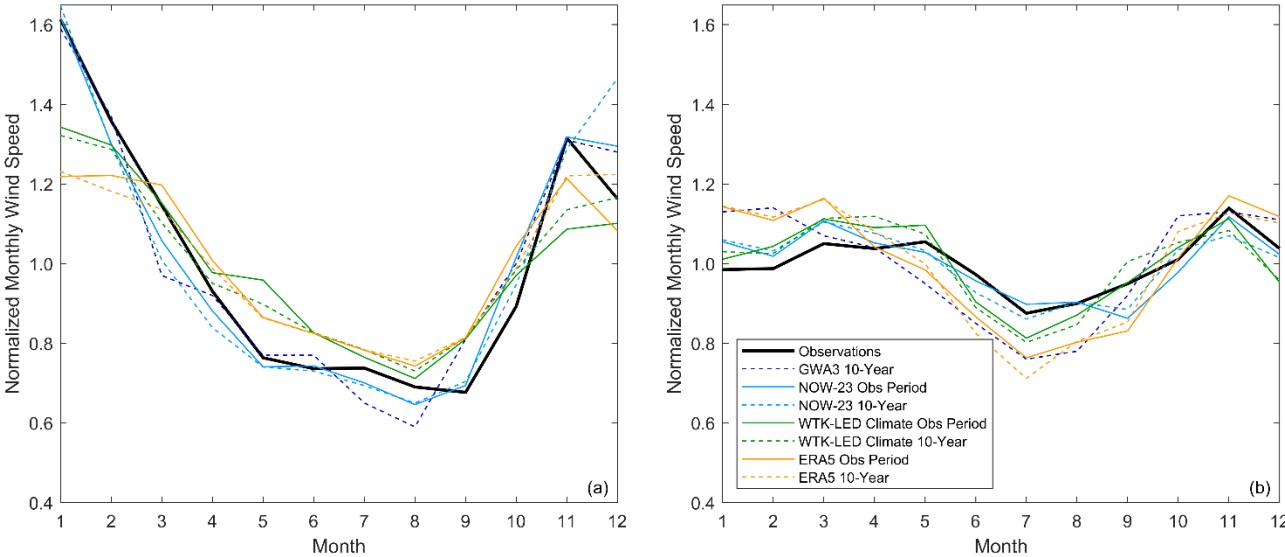

**Figure 9.** Observed and simulated normalized monthly wind speeds at the sites with (a) the most pronounced seasonal cycle (Oregon) and (b) the least pronounced seasonal cycle (Florida).

The same decade-long and observation-aligned approaches are utilised for studying the recent dataset representation of observed wind speed diurnal cycles (Figure 10). Understanding how the available wind resource changes throughout the day and night is important for distributed wind energy customers looking to reduce energy costs, particularly when time-of-use electricity pricing schedules are applied by local utilities. From a supply-and-demand standpoint, since diurnal peaks and troughs in electricity demand vary according to customer location and application (e.g., residential versus industrial facility

demand), a potential wind energy adopter will want to assess whether the times and degrees of wind generation will align with their energy needs. Finally, McCabe et al. (2022) highlight the importance of understanding diurnal (and seasonal) wind resource trends in the context of distributed wind complementarity with other energy technologies, such as solar energy. Distributed wind turbines and other energy technologies can be connected at the lower-voltage distribution level of an electricity grid to serve specific or local loads. In some instances, wind and other energy technologies may compete with

each other to provide electricity for a distributed load. In other instances, wind and other energy technologies may provide complementary solutions for the supply of clean electricity for distributed applications if they are generating on differing temporal schedules (McCabe et al., 2022).

    WTK-LED Climate produces the highest correlations with observations when considering the hourly trends in the wind speed (correlations = 0.87 using the full decade and the overlapping observational years). NOW-23 shows the next best

performance for representing diurnal trends in the wind resource (correlations = 0.86), followed by ERA5 (correlations = 0.78), and GWA3 (correlation = 0.62). Comparing the differences between the simulated and observed normalized hourly wind speeds for the best (WTK-LED Climate) and worst (GWA3) performing datasets in terms of diurnal representation reveals that GWA3 noticeably exaggerates the diurnal wind speed patterns for coastal sites along the Great Lakes, Gulf of Mexico, and Atlantic Ocean (Figure 11). In these three regions, GWA3 overestimates the observed normalized hourly wind

speeds at night and underestimates the observed normalized hourly wind speeds during the day at the majority of the observational sites. One suspected reason for the GWA3 challenges in representation of wind speed diurnal cycles is that the GWA3 diurnal (and seasonal and inter-annual) patterns are kept consistent across all dataset output heights from 10 m to 200 m, a scheme that is in contrast to studies that show observed differences (sometimes quite significant) between land-based diurnal wind speed patterns near the surface and those higher in the atmosphere near utility-scale wind turbine hub heights

(Wieringa, 1989; Barthelmie et al., 1996).

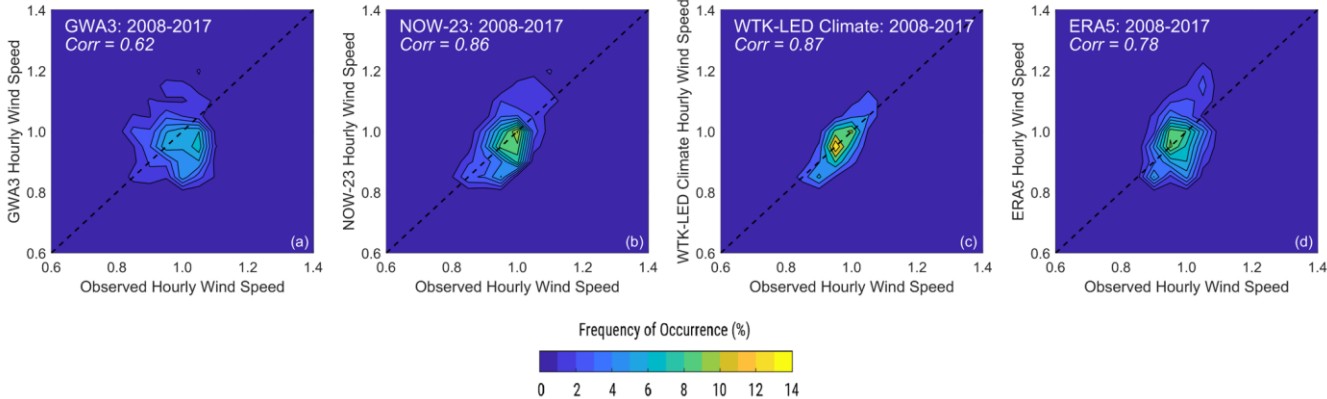

**Figure 10.** Observed and simulated normalized hourly average wind speeds across 23 coastal sites. Panels (a), (b), (c), and (d) depict the observed hourly average wind speeds versus the simulated hourly average wind speeds based on a decade (2008-2017) of data from GWA3, NOW-23, WTK-LED Climate, and ERA5, respectively. Correlations between NOW-23, WTK-LED Climate, and ERA5 with the wind speed observations during the temporal periods of the observations are identical to the decadal correlations.

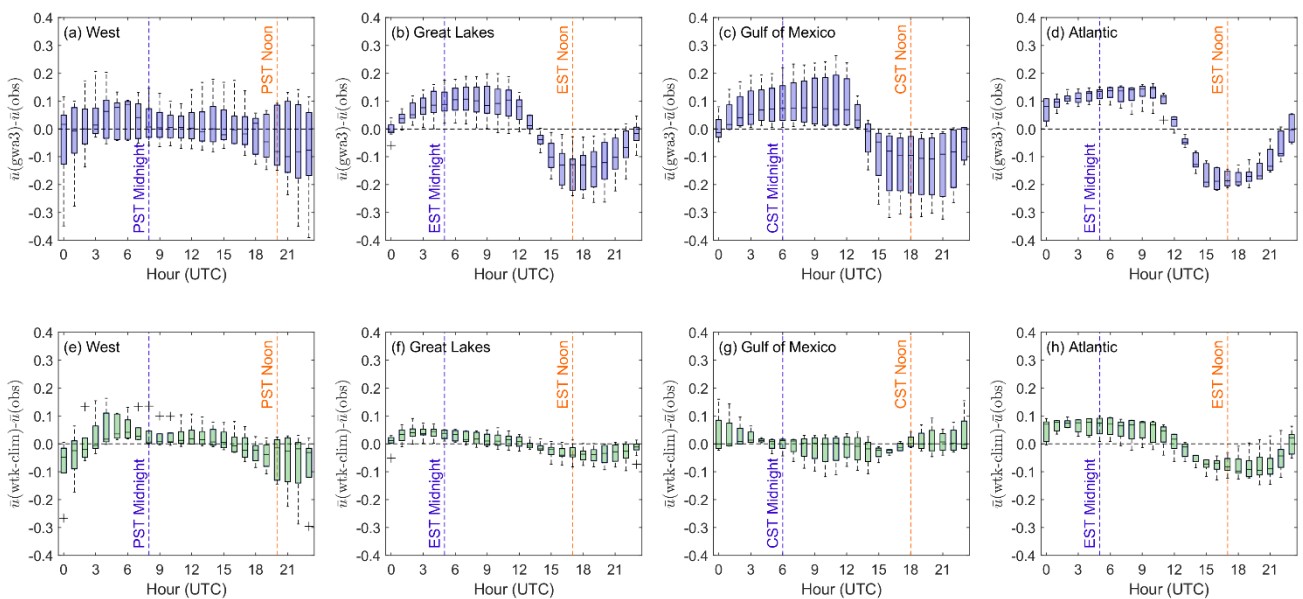

**Figure 11.** Difference between simulated and observed normalized hourly average wind speeds according to time of day. Panels (a) – (d) depict the differences in the hourly wind speeds between GWA3 and the observations. Panels (e) – (h) show the differences in the hourly wind speeds between WTK-LED Climate and the observations.

## 3.4 Inter-annual variability performance

Of the 23 coastal sites, 11 provide 6 or more years of observations, allowing for evaluation of the datasets for their representation of inter-annual fluctuations in the wind resource (Figure 2). According to region, six of the sites are along the Great Lakes, two sites are located in the West, two sites are on the Atlantic coast, and one site is along the Gulf of Mexico. Across the datasets, NOW-23 displays the best performance in representing the inter-annual variability in the wind resource,

with a correlation between the observed and simulated annual wind speeds of 0.81 (Figure 12). Following NOW-23 in inter-annual variability performance are ERA5 (0.77), GWA3 (0.74), and WTK-LED Climate (0.59).

The observations explain that a site in complex terrain near the Columbia River (approximately 15 km east of where the river empties into the Pacific Ocean) exhibits the highest inter-annual variability in the wind resource, with normalized annual wind speeds ranging from 0.89 to 1.08. An island off mainland Rhode Island has the lowest inter-annual variability,
with normalized annual wind speeds ranging from 0.98 to 1.02. While three of the four datasets (GWA3, NOW-23, and ERA5) correctly identify the Columbia River site as having the most pronounced inter-annual wind speed pattern, only NOW-23 identifies the Rhode Island site as having the least pronounced pattern. GWA3, WTK-LED Climate, and ERA5 instead identified a site off the coast of Lake Huron as having the least pronounced inter-annual wind speed pattern.

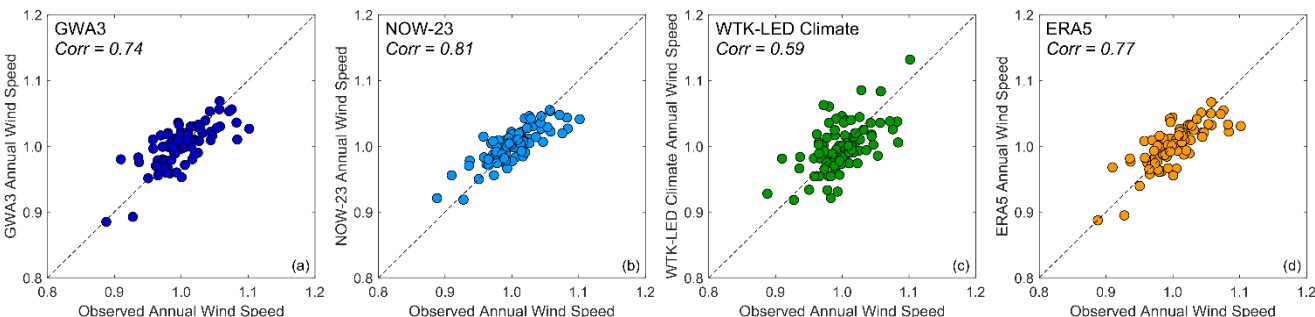

**Figure 12.** Observed and simulated normalized annual wind speeds across 12 coastal sites with 6 or more years of observations.

### 3.5 NOW-23 performance by regional dataset

Among the coastal sites, three are located within two unique NOW-23 domains, allowing for an additional, though limited in sample size, level of wind performance evaluation for this dataset. One site each in Connecticut and Rhode Island is located in both the North Atlantic (used in the analyses presented in Sections 3.1-3.4) and Mid Atlantic NOW-23 domains, while
another site in Florida is located in both the South Atlantic (used in the analyses presented in Sections 3.1-3.4) and Gulf of Mexico NOW-23 domains (Figure 1). The North and Mid Atlantic domains employ the same WRF setup with MYNN selected as the PBL scheme. The South Atlantic and Gulf of Mexico domains employ the same WRF setup with YSU chosen for the PBL scheme (Bodini et al., 2024).

In terms of annual average wind speed bias, the North Atlantic NOW-23 domain exhibits better performance for the
coastal observations than the Mid Atlantic NOW-23 domain. Specifically, the wind speed biases at the Connecticut and Rhode Island observational locations are approximately 0.2 m s$^{-1}$ lower using the North Atlantic domain (biases = 0.68 m s$^{-1}$ and 0.32 m s$^{-1}$, respectively) than the Mid Atlantic domain (biases = 0.91 m s$^{-1}$ and 0.49 m s$^{-1}$). At the Florida observational location, the South Atlantic and Gulf of Mexico domains perform similarly to each other, with annual average wind speed biases within 0.02 m s$^{-1}$ (biases = 1.36 m s$^{-1}$ and 1.34 m s$^{-1}$). At all three locations, the hourly correlations when using
different dataset domains are within 0.01 of each other (0.61 versus 0.62 for Connecticut, 0.75 versus 0.76 for Rhode Island,

and 0.54 versus 0.53 for Florida). The seasonal and diurnal trends in the NOW-23 wind speeds for the three sites are nearly identical, regardless of domain (Figure 13).

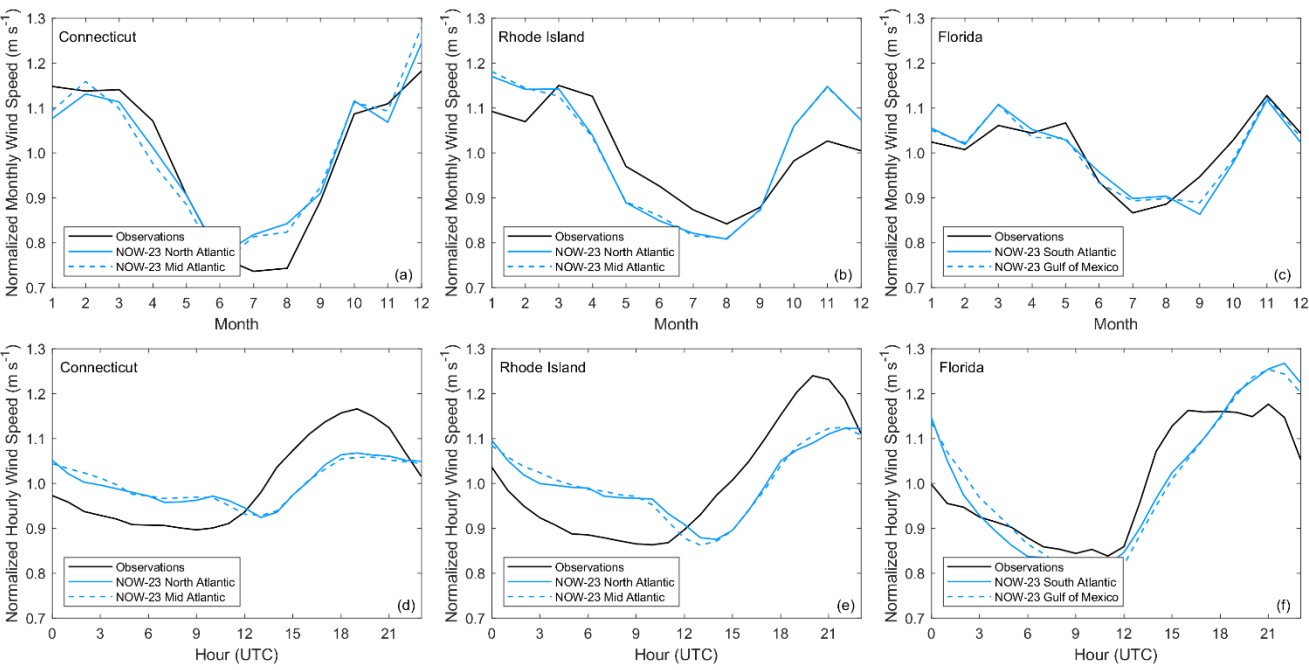

**Figure 13.** Observed and NOW-23 (a), (b), (c) monthly and (d), (e), (f) hourly wind speeds at observational locations in (a), (d)
Connecticut, (b), (e) Rhode Island, and (c), (f) Florida.

## 4 Discussion

Given that the significant time and economic investments involved with collecting pre-installation onsite wind resource measurements are often at odds with the timelines and available funds of communities, business owners, and residents interested in small or midsize wind turbine deployment, the free and user-friendly datasets evaluated in this article provide
crucial value in the wind speed estimates they provide. Additionally, the wind speed estimates for coastal communities can be adjusted using the validation results of this study. For example, because NOW-23 and WTK-LED Climate overestimate the observed annual average wind speeds at 96% of the study sites in this work, coastal users of these products might consider lowering their wind speed and energy production expectations.

Each of the recent wind datasets have strengths and limitations for coastal wind resource assessment. GWA3 provides the
lowest wind speed biases and relative errors for annual average wind speeds at the coastal observational sites (medians = 0.51 m s⁻¹ and 8.8%) (Table 3). NOW-23 is the optimal dataset for representing seasonal and inter-annual patterns in the coastal wind resource (median correlations = 0.92 and 0.81, respectively). WTK-LED Climate is the best performing dataset for diurnal cycle representation at the 23 coastal sites (median = 0.87).

In terms of limitations, GWA3 exhibits the greatest challenges at representing diurnal patterns (median correlation = 0.62), which could lead to challenges for a customer planning for energy coverage or offset around the clock using multiple distributed energy technologies. GWA3 also produces the lowest seasonal correlations compared to the other recent datasets, but still provides a relatively high level of accuracy (median = 0.83). WTK-LED Climate produces the highest annual average relative errors (median = 17.4%) and the lowest inter-annual correlations (median = 0.58). Providing accurate representations of the year-to-year variability in the wind resource is important for setting customer expectations for the wind energy that high, average, and low wind resource years will produce at their site.

**Table 3.** Summary of median wind speed performance metrics for GWA3, NOW-23, and WTK-LED Climate. The highest performing dataset for each evaluation metric is noted in bold.

| Evaluation Metric | GWA3 | NOW-23 | WTK-LED Climate |
|---|---|---|---|
| Annual Average Bias (m s$^{-1}$) | **0.51** | 0.98 | 0.88 |
| Annual Average Relative Error | **8.8%** | 16.1% | 17.7% |
| Seasonal Correlation | 0.84 | **0.92** | 0.89 |
| Diurnal Correlation | 0.62 | 0.86 | **0.87** |
| Inter-annual Correlation *(11 sites)* | 0.74 | **0.81** | 0.59 |

The evaluations in this work are limited by the spatial resolutions of the observations and by the temporal resolutions of some of the recent wind datasets. Future improvements to the evaluations of these and other wind datasets include validation across the wind profile using observations with multiple measurement heights. Additionally, for tools lacking hourly temporal resolution in their wind speed output or internal calculations, the implications of converting to energy estimates from low temporal resolution wind speed estimates must be evaluated. For higher temporal resolution products, such as NOW-23, performance at coastal locations in representing unique weather phenomena, such as low-level jets and sea breezes, can also be assessed. Finally, it is hoped that the validations provided in this work identify areas of future research for dataset developers, such as accuracy improvements for locations dominated by land-based flow and understanding of the NOW-23 discrepancies between 10 m and the rest of the wind profile.

**Appendix A**

Siting characteristics for the 23 coastal observations used for wind dataset validation are shared in Table A.1, including coordinates and measurement heights, wind roses, satellite imagery, general discussions of the land cover and nearby infrastructure, and citations to the original data sources. For each observational site, the 12 wind direction sectors used in this study are characterized as predominantly covered by land or water using the Global Land Cover and Land Use Change 2000-2020 (Potapov et al., 2022). For this analysis, an extent of 100 km from the observation location is utilised to capture both

onshore and offshore breezes (Gille et al., 2005; Viner et al., 2021). Each 100 km long directional sector, with its designation of predominantly open water, wetland, short vegetation, cropland, trees, or built-up coverage, is then weighted by the frequency of observed winds occurring for that directional sector.

**Table A.1.** Site characteristics of observations utilised in this study.

| Observation | Coordinates/ Measurement Height | Site Characteristics and Land Cover | Station ID (Source) | Terrain/Wind Rose | % of Winds According to Land Cover |
|---|---|---|---|---|---|
| **AL 1** | 30.228, -88.024<br>36 m | Long, low, narrow peninsula; few trees, large fort | FMOA1 (NDBC, 2024) |  | Open water 61%<br>Wetlands 4%<br>Short vegetation<br>Cropland<br>Forest 26%<br>Built-up 9% |
| **CA 1** | 33.733, -118.186<br>31 m | Shipping pier with high density infrastructure to the north and ocean to the south | PRJC1 (NDBC, 2024) |  | Open water 73%<br>Wetlands<br>Short vegetation<br>Cropland<br>Forest<br>Built-up 27% |
| **CT 1** | 41.306, -72.077<br>20 m | Lighthouse site with trees and infrastructure on mainland to the north and Long Island to the southwest; ~1 km to mainland | LDLC3 (NDBC, 2024) |  | Open water 42%<br>Wetlands<br>Short vegetation<br>Cropland<br>Forest 19%<br>Built-up 39% |

| | | | | | |
|---|---|---|---|---|---|
| **FL 1** | 27.933, -82.433<br>23 m | Surrounded by Tampa Bay and high density infrastructure | TPAF1<br>(NDBC, 2024) |  | Open water 33%<br>Wetlands 23%<br>Short vegetation 11%<br>Cropland<br>Forest<br>Built-up 33% |
| **IN 1** | 41.729, -86.912<br>21 m | Lighthouse at edge of pier; Lake Michigan to north, small city to south | MCYI3<br>(NDBC, 2024) |  | Open water 31%<br>Wetlands<br>Short vegetation<br>Cropland 63%<br>Forest<br>Built-up 6% |
| **LA 1** | 28.932, -89.407<br>20 m | Narrow peninsula into Gulf of Mexico; low-lying Mississippi delta basin; no infrastructure | PSTL1<br>(NDBC, 2024) |  | Open water 100%<br>Wetlands<br>Short vegetation<br>Cropland<br>Forest<br>Built-up |
| **MI 1** | 43.007, -82.422<br>27 m | Beach site south of Lake Huron; adjacent to river and suburban development | FTGM4<br>(NDBC, 2024) |  | Open water 20%<br>Wetlands<br>Short vegetation<br>Cropland 43%<br>Forest 10%<br>Built-up 27% |

| MI 2 | 43.228, -86.339 | Between Lake | MKGM4 |  | Open water 56% |
| | 24 m | Michigan and Lake | (NDBC, 2024) | | Wetlands |
| | | Muskegon; | | | Short vegetation |
| | | residential | | | Cropland 16% |
| | | development and | | | Forest 20% |
| | | forest | | | Built-up 8% |
| NJ 1 | 40.657, -74.065 | Lighthouse site in | ROBN4 |  | Open water 28% |
| | 21 m | New York Harbor; | (NDBC, 2024) | | Wetlands |
| | | surrounded by high | | | Short vegetation |
| | | density | | | Cropland |
| | | infrastructure in | | | Forest 13% |
| | | most directions | | | Built-up 59% |
| NY 1 | 42.494, -79.354 | Point extending | DBLN6 |  | Open water 49% |
| | 20 m | into Lake Erie; | (NDBC, 2024) | | Wetlands |
| | | nearby beach, | | | Short vegetation |
| | | forest, light | | | Cropland 4% |
| | | residential, | | | Forest 47% |
| | | industrial land | | | Built-up |
| | | cover | | | |
| NY 2 | 40.641, -74.162 | On Newark Bay; | MHRN6 |  | Open water 17% |
| | 46 m | surrounded by high | (NDBC, 2024) | | Wetlands |
| | | density | | | Short vegetation |
| | | infrastructure in | | | Cropland |
| | | most directions | | | Forest 19% |
| | | | | | Built-up 64% |

| OH 1 | 41.764, -81.281 | Lake Erie to the | FAIO1 |  | Open water 60% |
| | 21 m | north; mixed | (NDBC, 2024) | | Wetlands |
| | | beach, residential, | | | Short vegetation |
| | | and industrial land | | | Cropland |
| | | cover to the south | | | Forest 15% |
| | | | | | Built-up 25% |
| OH 2 | 41.629, -82.841 | Island in Lake | SBIO1 |  | Open water 29% |
| | 21 m[a] | Erie; nearby | (NDBC, 2024) | | Wetlands |
| | | forests, residential | | | Short vegetation |
| | | development, and | | | Cropland 59% |
| | | industrial | | | Forest |
| | | operations | | | Built-up 12% |
| OH 3 | 41.694, -83.473 | On Maumee Bay | THRO1 |  | Open water 19% |
| | 26 m | of Lake Erie; mix | (NDBC, 2024) | | Wetlands |
| | | of residential, | | | Short vegetation |
| | | forest, farm, and | | | Cropland 73% |
| | | industrial | | | Forest |
| | | | | | Built-up 8% |
| OR 1 | 45.558, -122.402 | Low lying | Troutdale |  | Open water |
| | 30 m | industrial location | (BPA, 2024) | | Wetlands |
| | | along Columbia | | | Short vegetation |
| | | River | | | Cropland 9% |
| | | | | | Forest 91% |
| | | | | | Built-up |

| RI 1 | 41.717, -71.345 | Lighthouse site; | CPTR1 |  | Open water 44% |
| | 21 m | mouth of | (NDBC, 2024) | | Wetlands |
| | | Providence River; | | | Short vegetation |
| | | high density | | | Cropland |
| | | residential on land | | | Forest 40% |
| | | | | | Built-up 16% |
| TX 1 | Proprietary | Low; near bay and | (DOE, 2024b) |  | Open water 66%[b] |
| | 60 m | Gulf of Mexico; | | | Wetlands |
| | | shrubland | | | Short vegetation 25% |
| | | | | | Cropland 9% |
| | | | | | Forest |
| | | | | | Built-up |
| VA 1 | 36.926, -76.007 | Point separating | CHYV2 |  | Open water 71% |
| | 28 m | Atlantic Ocean and | (NDBC, 2024) | | Wetlands 15% |
| | | Chesapeake Bay; | | | Short vegetation |
| | | nearby forested | | | Cropland 14% |
| | | state park and | | | Forest |
| | | commercial | | | Built-up |
| | | development | | | |
| WA 1 | 46.266, -123.877 | Ridgeline above | Megler RS |  | Open water 47% |
| | 53 m | Columbia River | (BPA, 2024) | | Wetlands |
| | | near Pacific Ocean | | | Short vegetation |
| | | outlet; trees and | | | Cropland |
| | | complex terrain | | | Forest 53% |
| | | | | | Built-up |

| | | | | | |
|---|---|---|---|---|---|
| **WA 2** | 46.422, -123.797<br><br>30 m | Ridgeline above Columbia River and Willapa Bay; trees and complex terrain | Naselle RS<br>(BPA, 2024) |  | Open water 39%<br><br>Wetlands<br><br>Short vegetation<br><br>Cropland<br><br>Forest 61%<br><br>Built-up |
| **WA 3** | 46.904, -124.105<br><br>27 m | Bay side; near marina, forested state park, and small town | WPTW1<br>(NDBC, 2024) |  | Open water 59%<br><br>Wetlands<br><br>Short vegetation<br><br>Cropland<br><br>Forest 41%<br><br>Built-up |
| **WI 1** | 47.079, -90.728<br><br>25 m | Forested island in Lake Superior<br><br>96% open water | DISW3<br>(NDBC, 2024) |  | Open water 60%<br><br>Wetlands<br><br>Short vegetation<br><br>Cropland<br><br>Forest 40%<br><br>Built-up |
| **WI 2** | 42.589, -87.809<br><br>20 m | Lake Michigan to the east, densely built city to west<br><br>56% open water, 41% built-up | KNSW3<br>(NDBC, 2024) |  | Open water 36%<br><br>Wetlands<br><br>Short vegetation<br><br>Cropland 35%<br><br>Forest<br><br>Built-up 29% |

[a] The SBIO1 anemometer height changed from 21 m to 24.3 m in September 2021 (NDBC, 2024). This change does not impact the results of this work, which covers 2008-2017, but is noted to avoid discrepancy concerns with station SBIO1's landing page.

[b] Wind direction observations are unavailable for the proprietary Texas location. Wind direction distributions for this site are taken from GWA3 and NOW-23 (the average of their distributions). The correlation of the wind direction distributions
between GWA3 and NOW-23 at this site is 0.97.

**Code and data availability**

All but one of the wind speed measurement datasets that support this study are publicly available. Measurements in the Pacific Northwest from the Bonneville Power Administration can be obtained at BPA (2024). Measurements from the National Data Buoy Center are sourced from NDBC (2024). The remaining measurement is proprietary, subject to non-
disclosure agreement, and has restricted access at DOE (2024b).

As for the simulated data, GWA3 is freely available at DTU (2024), NOW-23 is freely available at NREL (2024b), WTK-LED Climate is freely available at NREL (2024a), and ERA5 is freely available from ECMWF (2024). Data processing scripts are written in Matlab and are available from the contact author upon request.

**Author contributions**

Observational data management, software development, analysis, and manuscript preparation were performed by LS. Two of the datasets evaluated in this work were developed by JW, CJ, CD, RK, and NB. Dataset access was developed by DD, CP, and EY. Team management was performed by HT, DD, and CP. All authors contributed to the research conceptualization, manuscript edits, and technical review.

**Competing interests**

The contact author declares that none of the authors have any competing interests.

**Acknowledgements**

This work was authored by the Pacific Northwest National Laboratory, operated for the U.S. Department of Energy (DOE) by Battelle (contract no. DE-AC05-76RL01830). This work was authored in part by the National Renewable Energy Laboratory, operated by Alliance for Sustainable Energy, LLC, for the DOE under contract no. DE-AC36-08GO28308.
Funding was provided by the U.S. Department of Energy Office of Energy Efficiency and Renewable Energy Wind Energy Technologies Office. The views expressed in the article do not necessarily represent the views of the DOE or the U.S.

Government. The U.S. Government retains and the publisher, by accepting the article for publication, acknowledges that the U.S. Government retains a nonexclusive, paid-up, irrevocable, worldwide license to publish or reproduce the published form of this work, or allow others to do so, for U.S. Government purposes. The authors would like to thank Patrick Gilman and Bret Barker at the U.S. DOE Wind Energy Technologies Office for funding this research. The authors would also like to thank Suprajha Nagaraja Sudhakar, Sheri Anstedt, Rémi Gandoin, and one anonymous reviewer for thoughtfully reviewing the manuscript and providing helpful edits.

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
