# Peer review of "Performance of wind assessment datasets in United States coastal areas"

_Wind Energy Science, 2024_

## Referee Comment (RC1)

**Review of "Performance of wind assessment datasets in United States coastal areas"**
https://doi.org/10.5194/wes-2024-115.  Reviewer: R. Gandoin, C2Wind, Denmark.

Main text of the review

Thank you for a well written manuscript, the paper reads very well and provides useful comparisons and discussions.

Still, I think that some additional needs to be provided regarding three main topics:

- ➢ Measurement datasets: please consider adding additional information on the datasets, the reader needs to be able to contextualise the datasets, and if needed go back to the original time series for doing their own analysis.
- ➢ The definitions of the 10m winds in ERA5 and NOW-23, see points 2) and 3) below.
- ➢ When classifying sites using the land/sea ratio around the site, please consider the frequency of occurrence of onshore or offshore flow cases. As discussed in my comment in the pdf (page 7), a coastal site may experience onshore winds most of the time, making it more alike a onshore site wrt to model bias.

1) Could you please add additional information about the measurement datasets?

This should be, at the minimum, a table with a synthetic description of the measurement datasets with:

- ➢ coordinates
- ➢ explicit and univocal ID (so the user can easily find out which measurement dataset it is)
- ➢ data source with reference including link to the database

Ideally, this table would also include:

- ➢ a short description of the terrain (orography, roughness, obstacles nearby) and the measurement setup,
- ➢ a wind rose with a schematic coastline drawing/line
- ➢ an indication of whether the measurement location is in land or a sea model cell (or the value of the landmask, for ERA5)

Please consider as well providing:

- ➢ an energy-based metric about the type of most energetic flow case (onshore or offshore, see the main text of the review).
- ➢ the roughness length from the different modelling systems, at the measurement locations

2) Could you please discuss the definition of the 10m wind in ERA5?

The single levels 10m wind in ERA5 is, for onshore areas, not the model 10m wind. It is a diagnostic "WMO 10m wind" corresponding to a roughness of 3cm, see Section 3.10.2 of https://www.ecmwf.int/en/elibrary/79697-ifs-documentation-cy41r2-part-iv-physical-processes.

I think it is important to discuss the influence this may have on the discussion you are providing on model results comparisons.

2) Could you please discuss the question of the 10m wind in the NOW-23 MYNN

According to Section 2.6.1 of "A Description of the MYNN Surface-Layer Scheme" https://repository.library.noaa.gov/view/noaa/30605 the 10m wind in the MYNN is a neutral wind (for WRF version > WRF–ARWv4.0) is some specific flag is set to 1. The NOW-23 has been produced w WRF 4.2.1 (Bodini 2023), so this explains maybe the offset in the 10m wind value compared with the rest of the elevations.

I have myself seen an offset when looking at the NOW-23 dataset at the two floating lidars on the Pacific coast, see below comparisons I did based on data I received from NREL (the Vortex time series is a free 6 months long WRF run as well, from https://interface.vortexfdc.com/). These are offshore locations, so I am unsure what MYNN does there wrt the z0 value (Charnock?), but I see the difference between the 10 m and the otherwise expected value grows with stability.

It is worth double-checking if there anything here with the 10m wind from MYNN datasets that requires attention for this paper.

[revised manuscript text omitted]

---

## Author Comment (AC1)

Comments to the manuscript "Performance of wind assessment datasets in United States coastal areas"

General comments

This manuscript compares the existing wind speed datasets in the coastal regions of the United States by using measurement data at 23 points. The discussion is clear and the reviewer could not find any fundamental mistakes in their analysis. However, at the same time, the reviewer could not find any scientific or engineering value which is worth for this manuscript to be published as a research paper in Wind Energy Science journal. In other words, what are the findings in this work and what is the contribution of this work for wind energy community? The authors do not answer this most important question. As seen in the table 1, which is the summary of the datasets used in this work, the characteristics of the data are very different by datasets. For example, the method of downscaling is different, the temporal resolution is different and the spatial resolution is different, making it very difficult to investigate the cause of the error for each site. It seems that the discussion in this manuscript is very specific to the datasets and measurement sites, and it is hard to generalize the conclusion. Thus, the reviewer does not agree this manuscript to be published even with major revisions, unless substantial change are made with the introduction of the new viewpoints which brings more general conclusions.

Thank you so much for your time and effort in reading our manuscript. We apologize that we did not do a sufficient job of relating our results to how they will help the wind energy community. Our work provides quantification and analysis of the errors associated with multiple recently released wind resource datasets made available to wind analysts, which we feel aligns with Thematic Area 1 of the Wind Energy Science journal: Wind and the Atmosphere.

We are often asked by users of wind resource assessment tools and datasets, including manufacturers and installers of small and midsize wind turbines and potential distributed wind customers, about the accuracy of the products they use. The small and midsize distributed wind market typically does not have the time or monetary investment available to support gathering onsite observations and are thus dependent on wind resource datasets, such as those considered in this work, for pre-construction analysis.

Given this background, we completely agree with you that the characteristics of the datasets in this work are very different from each other, making it difficult to investigate the cause of the error for each dataset at each site. While we endeavor to investigate some of the discrepancies in dataset performance, such as the sensitivity of resultant annual average wind speed estimates based on which output heights a user selects, the main goal of this work is to share the dataset biases and relative errors, along with the accuracy of representing temporal trends in the wind resource. The datasets are recently released and easily accessible, therefore it is important to empower users with the level of confidence they should expect when working with them.

The following excerpts from the first and current drafts outline the goals of this work, the intended wind energy audiences we wish to support, and the usefulness of the findings:

Lines 14-24: "New models and tools are continually being developed in support of wind resource assessment, and three recent products are explored in this work for their performance in representing characteristics of the wind resource at coastal locations: the Global Wind Atlas 3 (GWA3), the 2023

National Offshore Wind data set (NOW-23), and the wind climate simulations that are a component of the Wind Integration National Dataset (WIND) Toolkit Long-term Ensemble Dataset (WTK-LED Climate). These relatively new products are freely available and user-friendly so that anyone from a utility-scale developer to a resident or business owner can evaluate the potential for wind energy generation at their location of interest.

    The validations in this work provide guidance on the accuracy of wind resource assessments for coastal customers interested in installing small or midsize wind turbines (≤ 1 MW in capacity) to support energy needs at the residential, business, or community scale, such as the island and remotely located participants of the U.S. Department of Energy's Energy Transitions Initiative Partnership Project."

Lines 51-53: "The sparsity of publicly-available observation data to support comprehensive wind resource assessment has driven the development of a variety of models, datasets, and tools over the last decade. Three datasets that were specifically developed to support the wind energy industry are evaluated in this work."

Lines 72-74: "The following analysis evaluates the performance of three recent wind assessment datasets in previously unvalidated locations along United States coastlines. The validation heights (20 m – 60 m) in this work support coastal communities interested in adopting small or midsize wind energy."

Lines 442-448: "Given that the significant time and economic investments involved with collecting pre-installation onsite wind resource measurements are often at odds with the timelines and available funds of communities, business owners, and residents interested in small or midsize wind turbine deployment, the free and user-friendly datasets evaluated in this article provide crucial value in the wind speed estimates they provide. Additionally, the wind speed estimates for coastal communities can be adjusted using the validation results of this study. For example, because NOW-23 and WTK-LED Climate overestimate the observed annual average wind speeds at 96% of the study sites in this work, coastal users of these products might consider lowering their wind speed and energy production expectations."

Lines 454-456: "GWA3 exhibits the greatest challenges at representing diurnal patterns (median correlation = 0.62), which could lead to challenges for a customer planning for energy coverage or offset around the clock using multiple distributed energy technologies."

Lines 457-460: "WTK-LED Climate produces the highest annual average relative errors (median = 17.4%) and the lowest inter-annual correlations (median = 0.58). Providing accurate representations of the year-to-year variability in the wind resource is important for setting customer expectations for the wind energy that high, average, and low wind resource years will produce at their site."

Per your concern that we did not adequately share our findings and the impact to the wind energy community, we have provided additional context to the discussion of the findings as follows:

Lines 267-273: "The recent datasets tend to overestimate the wind resource (GWA3 at 78% of the observation sites and NOW-23 and WTK-LED Climate at 96% of the observation sites). Disagreement between actual and predicted wind energy generation can lead to customer dissatisfaction and damage to the reputation of distributed wind as a viable energy resource, particularly in circumstances of overestimation. The findings of this work encourage users of GWA3, NOW-23, and WTK-LED Climate for coastal analyses to adjust their annual average wind speed and wind energy production expectations.

Additionally, the findings encourage the use of bias correction where possible, which can provide significant improvement to wind resource estimates (Wilczak et al., 2024)."

Lines 200-202: "The following sections compare recent wind assessment dataset performance at coastal sites versus the more established ERA5 in order to enable dataset users with the level of accuracy they can expect in representation of important pre-construction wind metrics, such as annual average wind speed and temporal trends in the wind resource."

Lines 471-473: "Finally, it is hoped that the validations provided in this work identify areas of future research for dataset developers, such as accuracy improvements for locations dominated by land-based flow and understanding of the NOW-23 discrepancies between 10 m and the rest of the wind profile."

Wilczak, J. M., Akish, E., Capotondi, A., and Compo, G.: Evaluation and Bias Correction of the ERA5 Reanalysis over the United States for Wind and Solar Energy Applications, Energies, 17, 7, 1667, https://doi.org/10.3390/en17071667, 2024.

Technical comments

Table 1.: What is the meaning of "Annual, seasonal, diurnal" in the temporal resolution rowof GWA3 and WTK-LED Climate? Does it mean annual average value per each year, seasonal average per season and diurnal average per a day are provided? But if so, the "temporal" resolutions is "once a day", isn't it?? (I mean annual or seasonal average value can be calculated from diurnal average data….) And the averaging time is different from the resolution and has to be specified separately. Anyway, more clarification needed

Thank you for your valuable suggestion to add clarity to the temporal resolution descriptions. We agree and have altered the description for GWA3 in Table 1 to read "Annual average wind speeds and normalized wind speed indices for establishing wind speed trends according to hour of day, month of year, and specific year in the 10-year coverage period."

Similarly, we have updated the description for WTK-LED Climate in Table 1 to read "Average wind speed by month and hour of day (12 x 24) for each year in the 20-year coverage period."

Equation 4.: The numerator of the right side of the equation looks like a ceiling function. But it does not make sense. Is it a simple bracket or absolute value?? From the following discussion, the relative error is always positive and the reviewer assumes this is an absolute value. But in that case, equation 4 has to be modified to the absolute value.

We are very grateful that you pointed this out! It is indeed intended to be an absolute value in the equation and we mistakenly used the ceiling brackets. We have corrected this in Equation 4:

$$relative\ error\ =\ 100\% * \frac{|\overline{u_{mod}} - \overline{u_{obs}}|}{\overline{u_{obs}}} \tag{4}$$

Figure 3: The difference of the different method to calculate the shear exponent is a little unclear. It's better to explicitly show by equation.

We appreciate this suggestion to add clarity to the different methods we used for calculating shear. While the basic equation for the shear exponent, defined in Equation 1, remains unchanged through this analysis, the averaging periods and wind speed output heights for the equation do change and we agree that the different scenarios can be challenging to keep track of. We've added the following table to the text to add clarity, per your helpful recommendation.

**Table 2.** Scenarios for determining the shear exponent for adjusting simulated wind speeds at dataset output heights to observational heights.

| Scenario | Description | GWA3 | NOW-23 | WTK-LED Climate | ERA5 |
|---|---|---|---|---|---|
| 1 | Analogous calculation using annual average wind speeds at output heights shared by all datasets (10 m and 100 m) | For each year, $\alpha = \dfrac{\ln(\overline{u_{100m}}/\overline{u_{10m}})}{\ln(100/10)}$ | For each year, $\alpha = \dfrac{\ln(\overline{u_{100m}}/\overline{u_{10m}})}{\ln(100/10)}$ | For each year, $\alpha = \dfrac{\ln(\overline{u_{100m}}/\overline{u_{10m}})}{\ln(100/10)}$ | For each year, $\alpha = \dfrac{\ln(\overline{u_{100m}}/\overline{u_{10m}})}{\ln(100/10)}$ |
| 2 | Calculation using annual average wind speeds at the nearest surrounding heights to each observation | For each year, $\alpha = \dfrac{\ln(\overline{u_{hi}}/\overline{u_{lo}})}{\ln(z_{hi}/z_{lo})}$ $z_{lo} = 10$ or $50$ m $z_{hi} = 50$ or $100$ m according to $z_{obs}$ | For each year, $\alpha = \dfrac{\ln(\overline{u_{hi}}/\overline{u_{lo}})}{\ln(z_{hi}/z_{lo})}$ $z_{lo} = 10, 20,$ or $40$ m $z_{hi} = 20, 40,$ or $60$ m according to $z_{obs}$ | For each year, $\alpha = \dfrac{\ln(\overline{u_{hi}}/\overline{u_{lo}})}{\ln(z_{hi}/z_{lo})}$ $z_{lo} = 10, 30,$ or $40$ m $z_{hi} = 30, 40,$ or $60$ m according to $z_{obs}$ | For each year, $\alpha = \dfrac{\ln(\overline{u_{hi}}/\overline{u_{lo}})}{\ln(z_{hi}/z_{lo})}$ $z_{lo} = 10$ m, $z_{hi} = 100$ m for all observations |
| 3 | Calculation at each dataset's highest temporal resolution using the nearest surrounding heights to each observation | For each year, $\alpha = \dfrac{\ln(\overline{u_{hi}}/\overline{u_{lo}})}{\ln(z_{hi}/z_{lo})}$ $z_{lo} = 10$ or $50$ m $z_{hi} = 50$ or $100$ m according to $z_{obs}$ | At each hour, $\alpha = \dfrac{\ln(u_{hi}/u_{lo})}{\ln(z_{hi}/z_{lo})}$ $z_{lo} = 10, 20,$ or $40$ m $z_{hi} = 20, 40,$ or $60$ m according to $z_{obs}$ | At each month/hour combination, $\alpha = \dfrac{\ln(\overline{u_{hi}}/\overline{u_{lo}})}{\ln(z_{hi}/z_{lo})}$ $z_{lo} = 10, 30,$ or $40$ m $z_{hi} = 30, 40,$ or $60$ m according to $z_{obs}$ | At each hour, $\alpha = \dfrac{\ln(u_{hi}/u_{lo})}{\ln(z_{hi}/z_{lo})}$ $z_{lo} = 10$ m, $z_{hi} = 100$ m for all observations |

Line 296-302: NOW-23 is based on different PBL scheme by different locations, right? However it does not justify to discuss the results as the difference of the PBL scheme as figure 8. They are based on different sites. This discussion is misleading and unacceptable.

Thank you for pointing out this concern. We have removed Figure 8 and the associated discussion.

Line 347-: What is the meaning to discuss the relative diurnal cycle?? The meaning of discussing the diurnal cycle for wind power application is not clear. The authors needs to clarify the justification of this discussion.

Thank you for the suggestion to add a discussion of why the wind resource diurnal cycle is important for wind energy customers. We have added the following discussion to Lines 371-382 to provide this important context:

"Understanding how the available wind resource changes throughout the day and night is important for distributed wind energy customers looking to reduce energy costs, particularly when time-of-use electricity pricing schedules are applied by local utilities. From a supply-and-demand standpoint, since diurnal peaks and troughs in electricity demand vary according to customer location and application (e.g., residential versus industrial facility demand), a potential wind energy adopter will want to assess

whether the times and degrees of wind generation will align with their energy needs. Finally, McCabe et al. (2022) highlight the importance of understanding diurnal (and seasonal) wind resource trends in the context of distributed wind complementarity with other energy technologies, such as solar energy. Distributed wind turbines and other energy technologies can be connected at the lower-voltage distribution level of an electricity grid to serve specific or local loads. In some instances, wind and other energy technologies may compete with each other to provide electricity for a distributed load. In other instances, wind and other energy technologies may provide complementary solutions for the supply of clean electricity for distributed applications if they are generating on differing temporal schedules (McCabe et al., 2022)."

McCabe, K., Prasanna, A., Lockshin, J., Bhaskar, P., Bowen, T., Baranowski, R., Sigrin, B., and Lantz, E.: Distributed Wind Energy Futures Study, National Renewable Energy Laboratory, Golden, CO (United States), NREL/TP-7A40-82519, https://doi.org/10.2172/1868329, 2022.

---

## Author Comment (AC2)

Review of "Performance of wind assessment datasets in United States coastal areas"

https://doi.org/10.5194/wes-2024-115. Reviewer: R. Gandoin, C2Wind, Denmark.

Main text of the review

Thank you for a well written manuscript, the paper reads very well and provides useful comparisons and discussions. Still, I think that some additional needs to be provided regarding three main topics:

- ➢ Measurement datasets: please consider adding additional information on the datasets, the reader needs to be able to contextualise the datasets, and if needed go back to the original time series for doing their own analysis.
- ➢ The definitions of the 10m winds in ERA5 and NOW-23, see points 2) and 3) below.
- ➢ When classifying sites using the land/sea ratio around the site, please consider the frequency of occurrence of onshore or offshore flow cases. As discussed in my comment in the pdf (page 7), a coastal site may experience onshore winds most of the time, making it more alike a onshore site wrt to model bias.

Thank you very much for your review and constructive suggestions for our work! We are grateful for your time and assistance. We have updated the manuscript based on your valuable feedback regarding the three main topics that required attention as outlined below.

1) Could you please add additional information about the measurement datasets?

This should be, at the minimum, a table with a synthetic description of the measurement datasets with:

- ➢ coordinates
- ➢ explicit and univocal ID (so the user can easily find out which measurement dataset it is)
- ➢ data source with reference including link to the database

Ideally, this table would also include:

- ➢ a short description of the terrain (orography, roughness, obstacles nearby) and the measurement setup,
- ➢ a wind rose with a schematic coastline drawing/line
- ➢ an indication of whether the measurement location is in land or a sea model cell (or the value of the landmask, for ERA5)

Please consider as well providing:

- ➢ an energy-based metric about the type of most energetic flow case (onshore or offshore, see the main text of the review).
- ➢ the roughness length from the different modelling systems, at the measurement locations

Thank you for this great suggestion! We have added Appendix A and Table A.1 to describe the coordinates, measurement heights, site characteristics, station IDs, original data sources, and wind roses for each observational site. Additionally, we incorporated your recommendation of defining an energy-based flow metric for each site. For this we characterized the wind rose sectors with a definition of predominantly land cover or water cover using the Global Land Cover and Land Use Change 2000-2020

(Potapov et al., 2022). We applied a radius of 100 km from each observation location for this analysis to capture both onshore and offshore breezes based on the works of Gille et al. (2005) and Viner et al. (2021) and then weighted the amounts of land and water coverage by the distribution of wind across the directional sectors. This information was also added to Table A.1. The only observation-related recommendations we did not include were the land vs sea model cell information and the model roughness lengths as these were not universally available for all the datasets considered in this work.

New text was added to the main body of the article (Lines 158-162) to discuss the energy-based metric as follows:

"When considering the distribution of flow direction within a 100 km radius to represent the extent of onshore and offshore breezes (Gille et al., 2005; Viner et al., 2021), the winds at 14 sites predominantly originate over land while the winds at 9 sites predominantly originate over water (Figure 2e, Appendix A) as determined by the Global Land Cover and Land Use Change 2000-2020 (Potapov et al., 2022) and the wind roses for each site."

A more robust discussion of the observations and the energy-based metric is provided in Appendix A (Lines 475-482):

"Siting characteristics for the 23 coastal observations used for wind dataset validation are shared in Table A.1, including coordinates and measurement heights, wind roses, satellite imagery, general discussions of the land cover and nearby infrastructure, and citations to the original data sources. For each observational site, the 12 wind direction sectors used in this study are characterized as predominantly covered by land or water using the Global Land Cover and Land Use Change 2000-2020 (Potapov et al., 2022). For this analysis, an extent of 100 km from the observation location is utilised to capture both onshore and offshore breezes (Gille et al., 2005; Viner et al., 2021). Each 100 km long directional sector, with its designation of predominantly open water, wetland, short vegetation, cropland, trees, or built-up coverage, is then weighted by the frequency of observed winds occurring for that directional sector."

Below is a capture of a portion of the new Table A.1:

**Table A.1.** Site characteristics of observations utilised in this study.

| Observation | Coordinates/ Measurement Height | Site Characteristics and Land Cover | Station ID (Source) | Terrain/Wind Rose | % of Winds According to Land Cover |
|---|---|---|---|---|---|
| AL 1 | 30.228, -88.024 36 m | Long, low, narrow peninsula; few trees, large fort | FMOA1 (NDBC, 2024) |  | Open water 61% Wetlands 4% Short vegetation Cropland Forest 26% Built-up 9% |
| CA 1 | 33.733, -118.186 31 m | Shipping pier with high density infrastructure to the north and ocean to the south | PRJC1 (NDBC, 2024) |  | Open water 73% Wetlands Short vegetation Cropland Forest Built-up 27% |
| CT 1 | 41.306, -72.077 20 m | Lighthouse site with trees and infrastructure on mainland to the north and Long Island to the southwest; ~1 km to mainland | LDLC3 (NDBC, 2024) |  | Open water 42% Wetlands Short vegetation Cropland Forest 19% Built-up 39% |

Per another of your helpful recommendations, we characterized the dataset errors according to the new energy flow metric instead of the simpler method we used earlier that only considered the land/water ratio and not the predominant wind directions:

Lines 289-294: "Of the 23 coastal sites in this analysis, 14 have wind flow distributions where most of the wind is arriving from land, while 9 have wind flow distributions where most of the wind is arriving from water (Figure 2e). Each region (Figure 1) is represented in both the water-dominant and land-dominant lists of sites according to flow. GWA3, NOW-23, WTK-LED Climate, and ERA5 perform notably better for the sites with water-dominant wind distributions, with median wind speed relative errors of 3.1%, 9.3%, 15.0%, and 8.6% respectively, than for the sites with land-dominant wind distributions, where the median relative errors are 15.8%, 25.5%, 18.3%, and 12.0% (Figure 6b)."

Lines 299-305: "All datasets follow the same trend of increasingly positive wind speed biases for land-dominant sites relative to the water-dominant sites. For GWA3, NOW-23, and WTK-LED Climate, the sites

with land-dominant wind distributions experience a greater degree of dataset overestimation, with median wind speed biases of 0.81 m s⁻¹, 1.33 m s⁻¹, and 0.98 m s⁻¹, respectively, while the median wind speed biases for the water-dominant sites are 0.05 m s⁻¹, 0.36 m s⁻¹, and 0.71 m s⁻¹ (Figure 1a). For ERA5, the degree of model underestimation is reduced for the land-dominant sites relative to the water-dominant sites, with median wind speed biases of -0.19 m s⁻¹ and -0.54 m s⁻¹."

[Figure]

**Figure 1.** Annual average wind speed (a) biases and (b) relative errors across the 14 coastal sites with land-dominant wind flow and the 9 coastal sites with water-dominant wind flow.

Gille, S., Llewellyn Smith, S., Statom, N.: Global observations of the land breeze, Geophysical Research Letters, 32(5), https://doi.org/10.1029/2004GL022139, 2005.

Potapov, P., Hansen, M. C., Pickens, A., Hernandez-Serna, A., Tyukavina, A., Turubanova, S., Zalles, V., Li, X., Khan, A., Stolle, F. and Harris, N.: The global 2000-2020 land cover and land use change dataset derived from the Landsat archive: first results. Front. Remote Sens. 3: 856903, https://doi.org/10.3389/frsen.2022.856903, 2022.

Viner, B., Noble, S., Qian, J-H., Werth, D., Gayes, P., Pietrafesa, L., and Bao, S.: Frequency and Characteristics of Inland Advecting Sea Breezes in the Southeast United States, Atmosphere, 12(8), 950, https://doi.org/10.3390/atmos12080950, 2021.

2) Could you please discuss the definition of the 10m wind in ERA5?

The single levels 10m wind in ERA5 is, for onshore areas, not the model 10m wind. It is a diagnostic "WMO 10m wind" corresponding to a roughness of 3cm, see Section 3.10.2 of https://www.ecmwf.int/en/elibrary/79697-ifs-documentation-cy41r2-part-iv-physicalprocesses.

I think it is important to discuss the influence this may have on the discussion you are providing on model results comparisons.

Thank you for the suggestion to add more information concerning ERA5 to this work, which we had neglected to do by instead focusing on the more recent datasets. First, we have improved Section 2.1 by including a discussion of ERA5 that incorporates your clarification on the 10 m wind along with the reference you kindly provided (Lines 129-133):

"ERA5 is a widely used global reanalysis model (Hersbach et al., 2020) in the wind energy community that began initial production in 2016. The single level ERA5 product outputs wind data at 10 m and 100 m above ground level (Table 1). The winds at the 10 m level are obtained via interpolation between the lowest model level and the surface and are corrected to align with open terrain observations. To adjust to the observations, the correction procedure for the ERA5 10 m winds involves an aerodynamic roughness length that is typical for open terrain with grassland (ECMWF, 2016)."

Second, we have added the roughness length consideration to the discussion of the dataset errors broken out according to land-dominant and water-dominant wind distributions on Lines 294-299:

"The significant decrease in dataset accuracy for land-dominant sites is likely due to a combination of challenges, including dataset representation of complex terrain (particularly for the western sites) and characterization of surface roughness length. Concerning the latter, the land-dominant sites tend to have wind flow distributions that favour cropland, forests, and built environments (Table A.1) which have greater roughness lengths than, for example, the open terrain grassland roughness length utilised for post-processing ERA5's 10 m single level output (ECMWF, 2016)."

ECMWF: IFS Documentation CY41R2 - Part IV: Physical Processes, https://doi.org/10.21957/tr5rv27xu, 2016.

3) Could you please discuss the question of the 10m wind in the NOW-23 MYNN

According to Section 2.6.1 of "A Description of the MYNN Surface-Layer Scheme" https://repository.library.noaa.gov/view/noaa/30605 the 10m wind in the MYNN is a neutral wind (for WRF version > WRF–ARWv4.0) is some specific flag is set to 1. The NOW-23 has been produced w WRF 4.2.1 (Bodini 2023), so this explains maybe the offset in the 10m wind value compared with the rest of the elevations.

I have myself seen an offset when looking at the NOW-23 dataset at the two floating lidars on the Pacific coast, see below comparisons I did based on data I received from NREL (the Vortex time series is a free 6

months long WRF run as well, from https://interface.vortexfdc.com/). These are offshore locations, so I am unsure what MYNN does there wrt the z0 value (Charnock?), but I see the difference between the 10 m and the otherwise expected value grows with stability.

It is worth double-checking if there anything here with the 10m wind from MYNN datasets that requires attention for this paper

[Figure]

We really appreciate the profiles you shared and the ideas on why we mutually are seeing divergent behavior between 10 m and the rest of the wind profile for NOW-23 domains using MYNN. We are interested in further exploration; however, Reviewer 2 was strongly against our analysis of NOW-23 performance according to PBL scheme for the reason that each site has a unique, and therefore incomparable, PBL scheme. They found the discussion and the former Figure 8 (the NOW-23 relative wind speed errors according to PBL scheme) misleading, and we have accommodated their concern by removing the analysis.

We have, however, expanded the detail in our near surface shear exponent graphic, Figure 4, to depict the NOW-23 domains in hopes of encouraging further research into this interesting profile behavior. Additionally, we noted this discrepancy in our discussion section to encourage it as an area for future investigative research:

Lines 471-474: "Finally, it is hoped that the validations provided in this work identify areas of future research for dataset developers, such as accuracy improvements for locations dominated

by land-based flow and understanding of the NOW-23 discrepancies between 10 m and the rest of the wind profile."

[Figure]

**Figure 4.** Shear exponents based on the lowest dataset output heights (x-axis) and 10 m and 100 m (y-axis) across 23 coastal sites from (a) GWA3, (b) NOW-23, and (c) WTK-LED Climate.

In addition to the main topics above, we are grateful for the notes you made throughout the body of the text and have addressed your suggestions as follows:

- Added the following to Lines 105-106: "Additionally, GWA3 provides Generalized Wind Climate files that include the wind speed and wind direction distributions for a number of roughness classes that a user can incorporate into WAsP."
- Updated the start year of ERA5 from 1950 to 1940 in Table 1.
- Added the following footnote to the ERA5 spatial resolution in Table 1: "The ERA5 data have been converted from the native reduced Gaussian grid to a regular latitude-longitude grid at 0.25° (Hersbach et al., 2020)."
- Added "for the single levels product" to the ERA5 output heights in Table 1.